# A ship-in-a-bottle quantum gas microscope setup for magnetic mixtures

Maximilian Sohmen[1,2,⋆,∘], Manfred J. Mark[1,2],
Markus Greiner[3] and Francesca Ferlaino[1,2]

**1** Institute for Quantum Optics & Quantum Information,
Austrian Academy of Sciences, 6020 Innsbruck, Austria
**2** Institute for Experimental Physics, Leopold-Franzens-Universität, 6020 Innsbruck, Austria
**3** Department of Physics, Harvard University, Cambridge, Massachusetts 02138, USA

⋆ maximilian.sohmen@i-med.ac.at

## Abstract

Quantum gas microscopes are versatile and powerful tools for fundamental science as well as promising candidates for enticing applications such as in quantum simulation or quantum computation. Here we present a quantum gas microscopy setup for experiments with highly magnetic atoms of the lanthanoid elements erbium and dysprosium. Our setup features a non-magnetic, non-conducting, large-working-distance, high-numerical-aperture, in-vacuum microscope objective, mounted inside a glue-free quartz glass cell. The quartz glass cell is enclosed by a compact multi-shell ferromagnetic shield that passively suppresses external magnetic field noise by a factor of more than a thousand. Our setup will enable direct manipulation and probing of the rich quantum many-body physics of dipolar atoms in optical lattices, and bears the potential to put exciting theory proposals – including exotic magnetic phases and quantum phase transitions – to an experimental test.



## Contents

---

∘ Current address: Institute for Biomedical Physics, Medical University of Innsbruck, Austria.

# 1 Introduction

## 1.1 Background

Understanding and control of the interplay between light and matter at the microscopic level has enabled revolutionary insights in fundamental physics and plays an increasing role for applications [1–5]. A pivotal advantage of such quantum optics approaches is the ability to custom-tailor quasi-pure model systems, which can be prepared, manipulated and probed with high fidelity while being near-perfectly isolated from environmental sources of noise. Quantum gas microscopes, in particular, enable the manipulation and imaging of individual, ultracold particles (typically: neutral atoms) pinned in a two-dimensional (2D) optical lattice potential, allowing to study quantum many-body physics on the single-particle level [6–8]. Requiring a high signal-to-noise ratio, site-resolved detection is usually based on fluorescence imaging (however, also Faraday imaging has been demonstrated [9]). To counteract recoil-heating of the pinned atoms by the scattered fluorescence photons (which could cause them to hop to nearby lattice sites), it is typically crucial to combine the imaging procedure with an in-trap optical cooling scheme.

The first quantum gas microscopes used bosonic alkali atoms ($^{87}$Rb) and went into operation in 2009/10 [10, 11]. They enabled studies of quantum phase transitions and the associated Higgs mode [12], particle correlations [13], quantum dynamics [14, 15] and other, similarly fundamental phenomena [6, 8]. About five years later, two groups demonstrated microscopes for the bosonic lanthanoid $^{174}$Yb [16, 17], whose complex electronic level structure promises opportunities for realising new quantum information protocols. Microscopy of alkali fermions – in contrast to the alkali boson $^{87}$Rb – initially proved challenging, since the small hyperfine splitting and low mass required the refinement of optical cooling procedures [6, 18]. The coming into operation of five different fermion microscopes in 2015 – using either $^6$Li [19, 20] or $^{40}$K [21–23] – marked the beginning of a long series of important results for Fermi systems, such as the direct observation of band [20] and (fermionic) Mott insulators [22, 24], antiferromagnetic ordering [25, 26], and many more [6, 8].

The vast majority of experiments with quantum gas microscopes has so far concentrated on atoms interacting via a short-range contact interaction. In such systems, atoms do not directly experience an energy shift depending on whether a neighbouring lattice site is occupied or not

– neighbour interactions are only introduced via a (weak) second-order tunnelling process, the so-called super-exchange interaction [27]. If, in contrast to purely contact-interacting systems, an additional interaction of long-range character – i.e., with an associated length scale similar to or larger than the lattice spacing – is present in a system, profound differences and new physics are to be expected [28, 29].

Currently, three different platforms are the main candidates for realising experiments with long-range interactions on optical lattices, and each of them has individual advantages and drawbacks. *First*, coupling optical-lattice ground-state atoms to Rydberg states induces a shift of the electronic energy levels, effectively dressing the atoms with a Rydberg character [30–32]. This results in an effective interaction potential between the atoms, whose strength and range can be controlled, e.g., via the detuning and intensity of the dressing laser, or the specific Rydberg state used. However, Rydberg admixtures have a limited lifetime and are highly susceptible to environmental stray electric fields [30]. *Second*, one can exploit the electric dipole-dipole interaction (DDI) between ground-state polar molecules [33–35]. Polar molecules, too, feature strong, long-range interactions which are – over a certain range – tunable via an external electric field, but this comes at the price of rather demanding molecule preparation schemes (many steps for association, cooling, collisional shielding, and others) as well as substantial particle losses due to complex, reactive collision processes [36]. *Third*, one can exploit the anisotropic and long-range magnetic DDI between atoms with a strong, permanent magnetic dipole moment such as chromium [37], dysprosium [38], or erbium [39]. Despite their lower interaction strength compared to Rydberg atoms and polar molecules, experimental realisations benefit tremendously from straightforward cooling and preparation schemes as well as long lifetime of samples of ultracold magnetic atoms.

Some direct effects of magnetic DDI on lattice physics have already been studied using conventional time-of-flight techniques [40–43]. Ultimately, however, it is very desirable to perform experiments with dipolar atoms on a lattice in a quantum gas microscope, where many phenomena are much more directly accessible. This has motivated several groups worldwide to start building quantum gas microscopes for dipolar atoms, including us as well as collaborators at Harvard, who in a recent preprint reported pioneering results [44].

## 1.2 Key features of erbium and dysprosium

All quantum gas microscopes are tailored specifically to the needs of their atomic species. This is necessitated by the demanding properties such a setup has to offer, for example high optical resolution or dedicated imaging and cooling schemes. Hence, let us first summarise some of the decisive features of erbium and dysprosium that distinguish them, e.g., from the alkali metals which predominate in quantum gas microscopy today.

*Magnetic moment.* Most importantly, of course, erbium and dysprosium feature large permanent magnetic dipole moments of $7\,\mu_{\mathrm{B}}$ and $10\,\mu_{\mathrm{B}}$, respectively, where $\mu_{\mathrm{B}}$ is the Bohr

Table 1: *Selected optical transitions of erbium (dysprosium), from broad to narrow, and examples for application.*

| $\lambda$/nm | $\Gamma/2\pi$ | | APPLICATION | REFS |
|---|---|---|---|---|
| 401 (421) | 29.4 (32.2) | MHz | Zeeman slowing, fluorescence imaging | [45, 46] |
| 583 (626) | 186 (135) | kHz | narrow-line MOT | [45, 46] |
| 741 (841) | 8 (1.8) | kHz | narrow-line MOT | [47, 48] |
| 1299 (1001) | 0.9 (11) | Hz | spin manipulation | [49, 50] |

magneton (cf. rubidium: $1\mu_B$). Importantly, the magnetic moment enters the DDI squared, giving about two orders of magnitude difference between erbium or dysprosium and the alkali atoms [51,52]. Further, most dipolar effects are determined by the ratio $\varepsilon_{dd} = a_{dd}/a$ between dipolar length, $a_{dd}$, and scattering length, $a$. Via $a_{dd} \propto m$ also the atomic mass $m$ enters [52], leading to much stronger dipolar effects for erbium/dysprosium than, e.g., chromium.

*Isotope options.* Both erbium and dysprosium posses a variety of bosonic as well as fermionic isotopes, which can be cooled and captured efficiently in a parallel magneto-optical trap [53] and of which several combinations have been brought to mixture degeneracy [54].

*Feshbach resonances.* The erbium and dysprosium isotopes offer dense Feshbach spectra with a comfortable number of broad resonances at easily accessible field strengths, favourable for contact interaction tuning [55–57] or molecule formation [58]. Also interspecies Feshbach spectra of several isotope combinations provide conveniently broad resonances [59].

*Electronic energy spectra.* Erbium and dysprosium possess a large number of electronic spectral lines (see, e.g., Fig. 1 in Ref. [52]), with a great variety of different widths, from broad to extremely narrow (see Table 1). From these transitions the most suitable ones can be picked for the desired task (such as cooling, imaging, shelving, etc.). Note that the broadest transitions of erbium and dysprosium are in the blue part of the visible spectrum, hence yield a high resolution according to the Abbe limit.

*Large mass.* Erbium and dysprosium are comparatively heavy elements (depending on the isotope, between 161 and 170 atomic mass units). Therefore, recoil velocities are exceptionally low for all optical transitions (in particular, compared to the light alkali elements).

*Large spin manifold.* One of the reasons for the large magnetic moments of erbium and dysprosium is the large angular momentum in the electronic ground states, with $J = 6$ ($J = 8$) for the bosonic isotopes of erbium (dysprosium), and $F = 19/2$ ($F = 21/2$) for fermionic erbium (dysprosium). The corresponding Zeeman and hyperfine states can be used to emulate spin Hamiltonians or to implement synthetic dimensions [60,61].

These features of erbium and dysprosium give opportunity for new techniques that we would like to try with our quantum gas microscope.

*Free-space imaging.* It has been proposed [62] that – similar to free-space imaging of a single atom released from an optical tweezer [63,64] – for erbium and dysprosium atoms on an optical lattice the combination of a broad electronic transition (i.e., high photon scattering rate) and large atomic mass (i.e., slow diffusion) might enable fast imaging protocols that work *without* optical cooling and pinning lattice. A first experimental demonstration of such a free-space imaging of a lattice gas has been reported very recently [44]. The ability to reliably reconstruct site occupations using such a fast, free-space imaging protocol can greatly simplify the detection scheme and offer a way to avoid the parity-projection problem almost all current quantum gas microscopes are facing [8]. In addition, it might prove useful for imaging of bulk systems with high resolution.

*Spin manipulation and detection.* The combination of large ground-state spin space and narrow optical transitions makes erbium and dysprosium ideal candidates for high-fidelity spin-state control [49,50]. This can be used to reliably prepare and probe specific states in the quantum gas microscope, to dynamically drive phase transitions [65], and to perform spin-selective imaging and shelving [66].

*DDI-mediated lattices.* The availability of many suitable optical transitions, combined with the light polarisation as an additional tuning parameter (due to a sizeable beyond-scalar polarisability) makes erbium and dysprosium promising candidates for implementing species-selective lattices (pinning for one species, invisible for the other) [67–71]. Due to interaction, the free species will see an effective periodic potential stemming from the pinned species. Since the pinned atoms can vibrate, the interaction-mediated periodic potential itself supports

phononic excitations – in contrast to a conventional, infinitely stiff optical lattice.[1] Compared to related works with purely contact-interacting atoms (see, e.g., Refs [73,74]), our system will allow to study and exploit the peculiarities brought about by the long-range and anisotropic nature of the DDI.

## 1.3 Potential research directions

There exists a wealth of promising research proposals that could be followed using a quantum gas microscope for dipolar atoms such as erbium or dysprosium, and yet further ones for a combination of two dipolar species. For instance, in lattice systems, the additional presence of the DDI requires an extension of the standard Bose- and Fermi-Hubbard models and lattice spin models. This gives rise to novel, qualitatively different quantum phases and transport dynamics. Here we list some examples of potential research directions that could in our opinion be particularly interesting to follow.

*Exotic ground states.* The long-range and anisotropic nature of the DDI dramatically changes the behaviour of bosonic as well as fermionic atoms in optical lattices. Dipolar bosons on a square lattice, for example, are expected to possess many-body ground-states resembling a charge-density wave. At half-filling, this can take the form of checkerboard or stripe configurations, and the phase diagram might host supersolid regions [28,29,51]. Changing the dipole orientation can drive transitions between these phases [44]; in close proximity of the transition, the existence of metastable emulsion phases has been predicted [75]. For spin-polarised dipolar fermions on a 2D lattice, in contrast, changing the dipole orientation is expected to give rise to a topological phase transition linked to the deformation of the Fermi surface, the so-called Lifshitz transition [76]. Taking the spin degree of freedom into account, erbium as well as dysprosium allows to implement a large number of different models, including spin-1/2 and spin-1 systems, with up to 20 (22) available spin states for fermionic erbium (dysprosium). A recent proposal suggests to use a 1D chain of spin-1/2 dipolar fermions to form a bond-order-wave phase, which is strongly correlated and topologically protected [77]. Other theory works have shown that also topological flat bands [78] or fractional Chern insulators [79] might be realisable. In principle, the whole class of systems described by spin Hamiltonians – like quantum magnetism models, spin liquids, and frustration – could be implemented using erbium and/or dysprosium atoms [80,81].

*Non-equilibrium dynamics.* Beyond the preparation and detection of exotic ground states, a quantum gas microscope is also well suited to study system dynamics. For example, it is expected that the DDI leads to cluster formation, where two or more atoms on neighbouring lattice sites are effectively bound together. These clusters can move around, but their speed is non-trivially dependent on the lattice geometry and system parameters [82,83]. Interestingly, such clusters can become fully localised without a need for disorder. Another phenomenon in reach is Levy flights – spin excitations that move through a sparse, randomly occupied lattice [84]. As a final example, it could be interesting to study the long-term dynamics of spin states within the Heisenberg XXZ model, where a peculiar difference in the equilibrium distribution is expected between scenarios with half-integer compared to integer-spin manifolds [85].

In the following, we describe the quantum gas microscope that we have designed as an addition to our erbium-dysprosium quantum mixture experiment. The microscope is housed in a separate ultra-high vacuum (UHV) section and has been assembled, evacuated, baked, tested, and finally attached to the existing apparatus through a mechanical UHV gate valve.

---

[1]Note that following a different approach, very recently an optical lattice supporting phonons was realised using a confocal optical resonator [72], shedding first light on the rich physics of elasticity in quantum solids.

## 2 Microscope objective

The most important component of the Er-Dy quantum gas microscope is its imaging objective. In the following, we will first discuss design options and motivate our choice. Next, we detail our design's optical and mechanical properties, and finally present the measured performance.

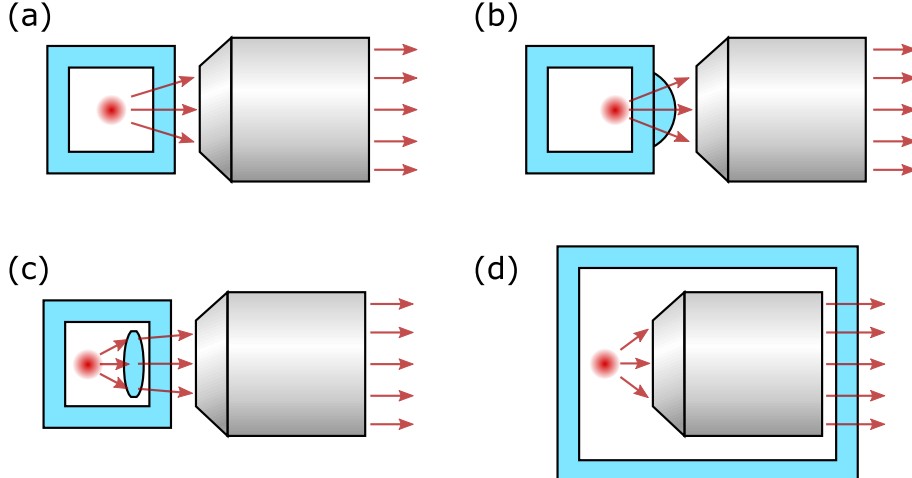

Figure 1: *Types of quantum gas microscopes.* (a) All optics outside vacuum chamber, (b) solid-immersion lens (SIL) close to sample, (c) optics partly in vacuum, (d) entire optics in vacuum.

Table 2: *Design types of quantum gas microscopes.* WD = working distance, NA = numerical aperture, $\lambda_l$ = lattice laser wavelength, UHV = ultra-high vacuum.

| TYPE | PRO | CONTRA | EXAMPLES |
|------|-----|--------|----------|
| (a) | easy alignment<br>long WD | limited NA | [11, 17, 20, 23, 86] |
| (b) | high NA<br>mech. rigid | short WD<br>$\lambda_l$ fixed by coating | [16, 19, 22] |
| (c) | moderate NA<br>moderate WD | difficult alignment<br>relative vibrations | [87–89] |
| (d) | high NA<br>easy alignment<br>long WD | UHV risks | [90–93] |

### 2.1 Initial options and considerations

Quantum gas microscopes are highly complex machines, often operating at the edge of what is technologically possible. Therefore, the optical design typically needs to be carefully tailored to the experimental needs. For the Er-Dy experiment, we identified the following key requirements for the imaging system:

1. a high numerical aperture (NA) > 0.8 to be able to (i) resolve lattices of small spacing ($\leq 532\,$nm) and for (ii) a high fluorescence photon collection rate;

2. a large working distance (WD) on the order of millimetres to give optical access from the side, to grant full freedom of (transverse) lattice laser wavelengths $\lambda_l$, and to avoid possible effects of close surfaces on the dipolar atoms;

3. use of non-magnetic and non-conducting materials to avoid magnetisation and eddy currents;

4. near achromaticity at the imaging wavelengths of erbium and dysprosium.

When a sample inside a UHV environment is imaged, the vacuum window necessarily becomes part of the light path and needs to be considered in the optical layout. The aberrations introduced by a plane-parallel glass plate are mainly spherical and chromatic and scale with the thickness of the plate [94]. Different quantum gas experiments have found different solutions for dealing with these aberrations, which can be broadly grouped in four categories as sketched

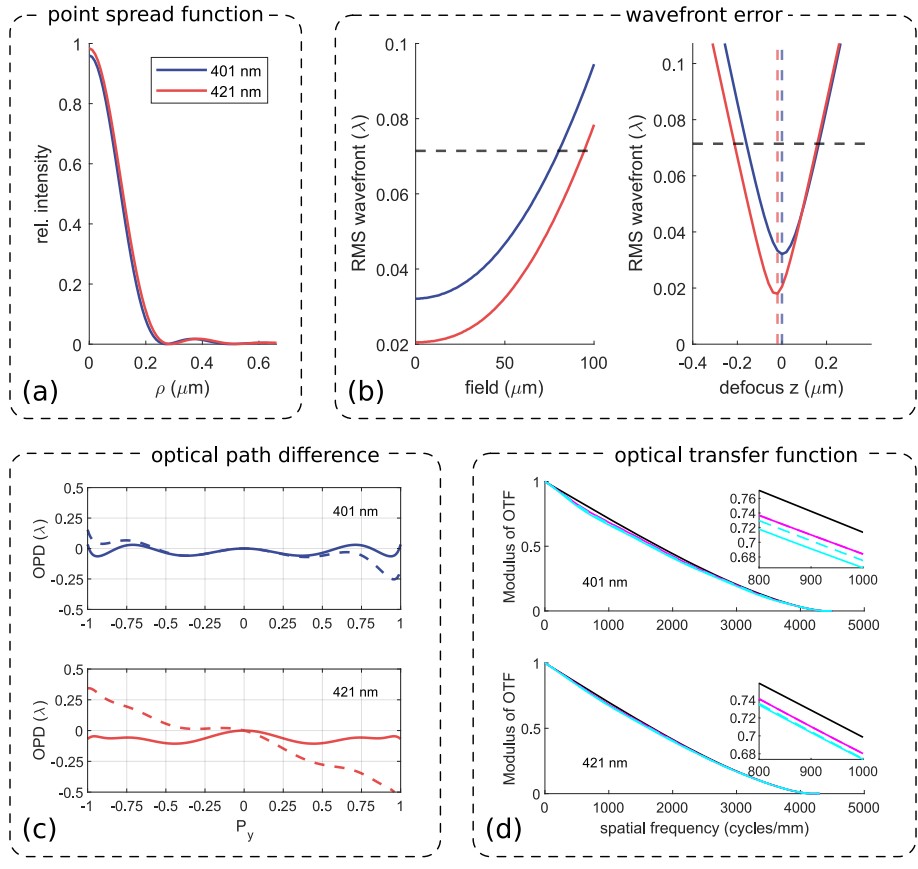

Figure 2: *Key characteristics of our microscope objective at the erbium and dysprosium imaging wavelengths.* (a) On-axis point-spread function (PSF) vs radial coordinate $\rho$ (in object-space units); the value at $\rho = 0$ equals the Strehl ratio. (b) root-mean-square (RMS) wavefront error vs lateral field (*left*) and vs focus position (*right*). Dashed horizontal lines give the diffraction limit. (c) Tangential fan of optical path difference (OPD) compared to chief ray vs normalised pupil coordinate $P_y$. Solid (dashed) lines correspond to zero ($40\,\mu$m) lateral field along $y$. (d) Optical transfer function (OTF); the solid black line gives the diffraction limit. Magenta (cyan) corresponds to to zero ($40\,\mu$m) lateral field along $y$. Solid (dashed) lines correspond to tangential (sagittal) direction. *Insets:* A zoom-in for intermediate spatial frequencies. Our calculations were carried out using Zemax Optic Studio 16.

in Fig. 1. Table 2 gives a brief overview over the most important properties and limitations of these approaches.

(a) *Optics outside vacuum.* This straight-forward option usually features a long WD of some to tens of millimetres which, for reasonable optics diameters, limits the NA to below $\sim 0.7$. The usual approach is to use a custom objective carefully corrected for a window of some millimetres thickness [11, 17, 20, 86]. However, the use of a commercial objective in combination with a very thin ($< 1\,\text{mm}$) and – due to bending under vacuum – small-diameter window has also been demonstrated [23].

(b) *Solid-immersion lens (SIL).* Lenses in shape of a truncated sphere close to the sample increase the NA [95]. Hemispherical SILs offer an enhancement factor equal to their refractive index, $n$, and can be mounted in vacuum [10] or be optically contacted to the glass window [16, 19, 22]. Notably, a window that is part of the SIL does not add aberrations. SILs with Weierstraß truncation offer an even higher NA enhancement by $n^2$ [93, 96, 97]. However, SILs have a very small (typically micrometre-scale) working distance, necessitating sophisticated transport strategies to bring the atoms into focus. Additionally, most often horizontal as well as vertical lattice beams have to be reflected off the front surface, posing many constraints for the optical coating, whose performance can quickly become limiting.

(c) *Optics partly in vacuum.* A first lens inside vacuum can reduce the marginal ray angles and hence aberrations by the following window. Imaging systems of this type have been demonstrated with moderate NAs around 0.5 [87–89]. Their disadvantage is that the relative alignment of in- and ex-vacuum components is absolutely crucial and that great care must be taken to prevent mechanical vibrations or long-term drifts.

(d) *All optics in vacuum.* In-vacuum objectives are conceptually simple and allow for both a millimetre-level working distance and high NA. If the objective is at infinite conjugation, the glass window does not add aberrations. Despite these advantages, this design is less frequently encountered in cold-atoms experiments, since the objective must be UHV-compatible and the UHV chamber must be large. Nevertheless, objectives of this kind have successfully been demonstrated in metal chambers [90, 92], in a glued glass cell [91] and, very recently, in a quartz glass cell similar to ours [93] – a parallel development for a neighbouring setup in Innsbruck.

For our objective, option (a) was rejected since it can hardly deliver the targeted NA and resolution. The SIL design (b) – due to its short WD and coating limitations – seemed incompatible with the number of different wavelengths that need to be directed onto (multiple lattices, cooling, spin manipulation, etc.) and collected from (broad-angle fluorescence) the erbium *and* dysprosium atoms. Option (c) seemed risky since, at the level of optical resolution aimed for, tolerances on alignment imperfections are small, and drifts or relative vibrations could have proven fatal for the optical performance. We hence opted for an in-vacuum objective (d), which – in combination with a glue-free glass cell – reminded us of the challenge to build a ship in a bottle.

## 2.2 The Er-Dy objective design

In this section, we present the high-NA in-vacuum objective for microscopy of erbium and dysprosium atoms which we have developed together with a manufacturer.[2] It was designed for achromatic performance on the broad, blue imaging transitions of erbium and dysprosium at 401 and 421 nm wavelength, respectively, as well as for 633 nm, which is the alignment wavelength of the manufacturer and, by coincidence, close to the red dysprosium transition at 626 nm. Let us highlight that while our blue imaging lines are beneficial in terms of the Abbe resolution limit, many standard optical elements do not perform well in this part of the

---

[2]Special Optics, Inc., NJ/USA.

spectrum. Also for the optical design of our microscope, the blue regime presented challenges: Here, on the one hand, refractive index dispersion is typically steep, and on the other hand, many standard optical glasses show non-negligible absorption, greatly reducing the choice of glasses. Note that absorptive losses would be particularly problematic for free-space single-atom fluorescence imaging, where often only a few tens of photons per atom are detected [44, 64]).

Table 3: *Important design values of the Er-Dy objective.*

| QUANTITY | UNIT | DESIGN VALUE | |
|---|---|---|---|
| eff. focal length | mm | 20.0 | |
| total length | mm | 70.0 | |
| depth of focus | µm | ±0.25 | |
| working f-number[a] | | 0.56 | |
| object-space NA | | 0.89 | |
| | | 401 nm | 421 nm |
| object-space Airy radius | µm | 0.27 | 0.29 |
| wavefront error[a] peak-valley | $\lambda$ | 0.098 | 0.11 |
| wavefront error[a] RMS | $\lambda$ | 0.032 | 0.021 |
| Strehl ratio[b] | | 0.96 | 0.98 |
| Ø diff.-lim. FOV | µm | 160 | 180 |

[a] Evaluated over full pupil.

[b] On axis (zero field).

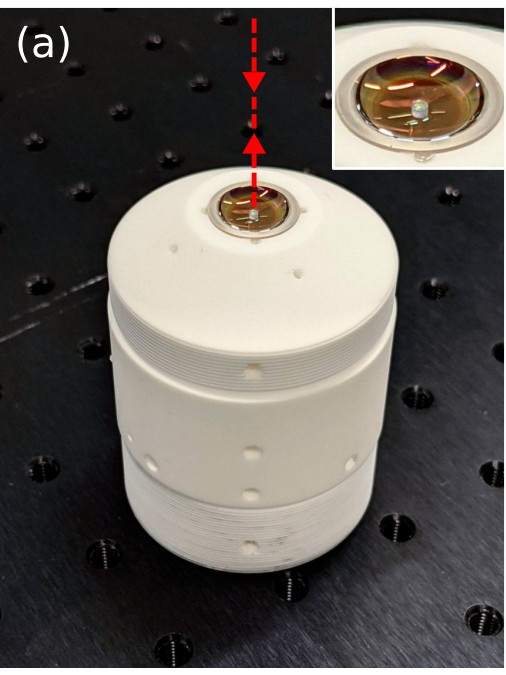
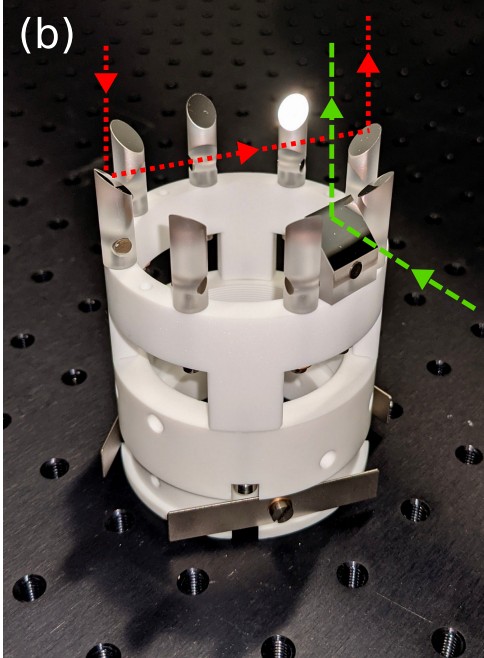

Figure 3: *In-vacuum optics.* (a) The microscope objective with the miniature lattice mirror (*inset*). The red dashed arrow indicates a vertical lattice beam. (b) The objective mount with crown mirrors. Round mirrors serve to reflect light onto the atoms and back (red dotted arrows), the rectangular mirror protects the objective from the divergent transport beam (green dashed arrows).

Our objective consists of five singlet lenses of different glasses with sufficient transmission for blue light; cemented elements have been avoided for the risk of outgassing. The objective tube and all lens retaining mechanics are fabricated from machinable ceramics. All volumes inside the housing and in between lenses are vented to avoid virtual leaks under UHV. The objective's optical design values are summarised in Table 3. In Fig. 2 (a–d), we present calculated characteristics[3] of our objective at the main imaging wavelengths (401 nm and 421 nm). In particular, (a) the PSF confirms an Airy radius of $< 0.3\,\mu m$, (b) the root-mean-square (RMS) wavefront error suggests a diffraction-limited lateral field of $> 70\,\mu m$, an expected chromatic focal shift below $0.1\,\mu m$, and a depth of focus of about $\pm 0.2\,\mu m$, (c) the optical path difference (OPD) is flat over the entire pupil, and (d) the optical transfer function (OTF) looks as expected for a well-corrected optical system. These simulation results will be discussed together with experimental measurements in Section 2.3.

A custom flat miniature dielectric mirror[4] of 1.5 mm diameter is glued centrally onto the concave objective front lens, see Fig. 3 (a). Alignment and glueing[5] were carried out by the objective manufacturer. This miniature mirror blocks only $\sim 3\,\%$ of the solid angle covered by the objective lens (NA = 0.89), so hardly affects the number of collected photons, and will serve to reflect off the vertical lattice beams ($\lambda_l = 1550\,nm$). This fixes the lattice position relative to the objective and will help to reduce drifts and vibrations, thus facilitate keeping the atoms in focus.

The objective is mounted in a home-built ceramics mount, shown in Fig. 3 (b). For transverse positioning, this ceramics mount features three titanium flat springs around its perimeter which keep it centred inside the glass cell bottom tube. For axial positioning, the ceramics mount rests on three ruby balls[6] which sit on the bottom fused-silica window of the glass cell (cf. Section 3.1). The individual pieces of the mount are assembled using vented titanium screws and beryllium-copper disc springs which take up mechanical stress, such as upon temperature changes during bake-out.

The top rim of the mount bears an arrangement of custom solid-quartz-glass mirrors[7] with protected metallic-silver coating, which we colloquially refer to as the 'crown mirrors'. Eight crown mirrors (elliptic in Fig. 3 b) are in staggered alignment with the side windows of the glass cell and will serve to reflect laser beams onto the atoms, entering and exiting through the large top window. One crown mirror (rectangular in Fig. 3 b) protects the objective from being hit by the divergent, high-power optical-transport beam (when its focus is away from the glass cell). We chose the angle of the crown mirrors such that normal incidence on the top window (and, thus, any standing-wave backreflection) is avoided.

We highlight that our microscope cell and interior are exclusively built from materials which are *non-magnetic* and – besides small parts like titanium screws and beryllium-copper disc springs – *non-conducting*. We thus maximally avoid magnetisation effects and eddy currents that could deteriorate the magnetic environment close to our magnetic atoms as well as limit field switching times.

## 2.3 Optical performance

The performance of the microscope objective was tested by imaging the tip of a scanning near-field optical microscopy (SNOM) fibre[8] onto a CCD camera with 50× magnification [97,98]. The SNOM fibre tip has a nominal aperture of 50 to 100 nm and therefore can be used as

---

[3] Modelled using Zemax Optic Studio 16.
[4] Optics Technology, Inc., NY/USA.
[5] Optocast 3415, Electronic Materials, Inc., CO/USA.
[6] Edmund Optics, Inc., NJ/USA.
[7] Optico AG, Switzerland.
[8] MF001, Tipsnano OÜ, Estonia.

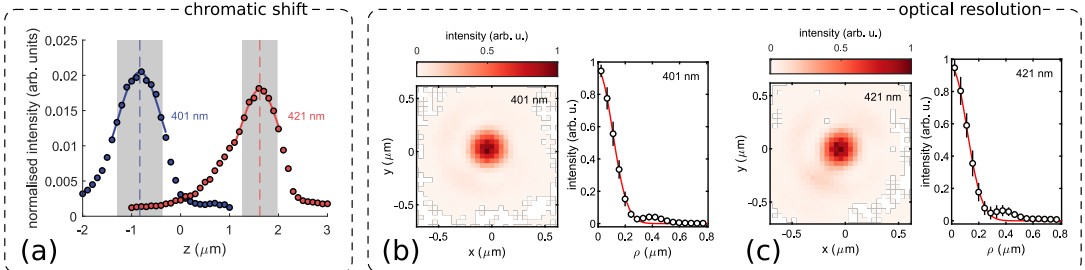

Figure 4: (a) *Chromatic focal shift.* Normalised intensity vs on-axis position ($z$), for for 401 nm (blue) and 421 nm (red). Solid lines are Gaussian fits to the maxima; dashed lines represent the centres, grey shadings the standard deviations. The distance between the maxima is around 2.4 µm. (b, c) *Optical resolution.* Images of a scanning near-field optical microscopy (SNOM) fibre tip (*respective left*) at 401 nm (b) and 421 nm (c). Distances ($x, y$) are in object-plane units. The azimuthally averaged intensity profiles (*respective right*) are plotted vs the radial coordinate ($\rho$). The red lines are Gaussian fits.

a good approximation of a point source. At constant magnification, the peak intensity in a power-normalised image is proportional to the Strehl ratio [99]. Figure 4 (a) shows the peak normalised intensity when moving the SNOM fibre tip along the optical axis using a piezo actuator. We observe a clear maximum – corresponding to the respective focus position – for each of the two investigated wavelengths (401 and 421 nm). The axial distance between these maxima, the chromatic focal shift, is $\sim 2.4$ µm. The objective is therefore close to, but not fully achromatic, as initially targeted. According to the manufacturer, this is most probably due to insufficient accuracy of their prior knowledge of refractive indices of the lens glasses (they had to be extrapolated down to 401 nm wavelength, where the dispersion is steep). For imaging of only one species per time, this does not pose a problem, since the shift can be corrected by re-adjusting the camera position. To be able to image both species in a single experimental run, the chromatic shift needs to be corrected for. Possible experimental solutions include (i) using a dichroic mirror to separate the beam paths and image the two species on separate cameras, or to image both species shortly after each other in combination with either (ii) an adaptive optical element, such as a fast focus-tunable lens [100], (iii) an imaging lens on a fast translation stage to dynamically adjust the focus position, or by (iv) dynamically shifting the vertical lattice position between the respective focal planes.

Figure 4 (b) shows images close to the foci [i.e., around the maxima in Fig. 4 (a)] at 401 and 421 nm, respectively, as well as azimuthally averaged and fitted spot profiles. These measurements give an upper bound on the optical resolution $d_0$ according to the Rayleigh criterion of

$$d_0 = \begin{cases} 0.29(1)\,\mu\text{m} & \text{at} \quad \lambda = 401\,\text{nm}, \\ 0.30(1)\,\mu\text{m} & \text{at} \quad \lambda = 421\,\text{nm}, \end{cases}$$

close to the values predicted by our simulations (Table 3).

The fluorescence from the atoms is collimated by the microscope objective (focal length $f = 20$ mm), passes the vacuum window, and needs to be re-focused (focal length $f'$) on a camera chip to form an image. For Nyquist sampling, a sufficient image magnification $M$ is needed. Aiming for, e.g., a sampling of five pixels per lattice site with a sensor pixel size of $d_{\text{px}} = 16.5$ µm (a typical value for EMCCD cameras), a lattice constant of 0.266 µm, we would require $M \gtrsim 310$ and $f' \gtrsim 6.2$ m. For larger lattice spacings or smaller pixel size these numbers are smaller, but still likely on the order of metres. Such long light paths would naturally suffer from stability problems caused by mechanical vibrations or air flow. Using numerical methods

we have identified stock lenses[9] that can be combined into a telefocus system which has a large effective focal length ($\sim 6.2$ m) but a small physical length ($\sim 1.1$ m) and is fully achromatic at 401 and 421 nm. This telefocus system has not yet been tested in combination with the microscope objective (now under vacuum); once we will have atoms loaded into the optical lattice, we will fully characterise the complete imaging system for reliable interpretation of images.

# 3 Vacuum integration

Experiments with ultracold atoms have to be conducted under UHV on the order of $10^{-11}$ millibar to isolate the samples from the environment and permit sufficiently long lifetimes. The basic design of the our microscope UHV setup consists of a quartz glass cell – housing the objective – attached to a stainless-steel[10] tube cross which connects to vacuum instrumentation and the main experiment.

## 3.1 Quartz glass cell

The microscope objective is mounted inside a quartz glass cell (Fig. 5). The quartz glass cell (compared to, e.g., a metal vacuum chamber) brings three main advantages: First, superb vacuum quality, second, it is inherently non-magnetic and non-conducting, third, it offers maximum optical access. Our custom-manufactured[11] glass cell consists of a hollow, octagonal quartz glass corpus with one 3" window attached to the top and seven 1" windows attached to side borings using a glass-frit bonding technique. Fused silica (i.e., high-purity synthetic quartz glass) was our window material of choice due to its low light absorption down to below 400 nm, and small thermal lensing effects even at high light intensity [101]. On the inside, our

---

[9]Hastings achromatic triplet ($f = 40$ mm) and achromatic doublet ($f = 1000$ mm), Thorlabs, Inc., NJ/USA.
[10]316LN and 316L (AISI classification) for low relative magnetic permeability.
[11]Precision Glassblowing of Colorado, Inc., CO/USA.

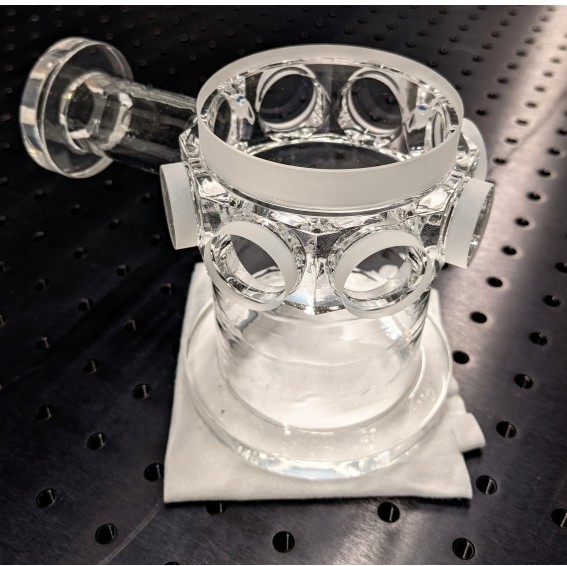

Figure 5: *The microscope cell.* Manufactured from quartz glass (body) and fused silica (windows). The polished flat lips on left and bottom tube are for indium sealing. The viewports feature a broadband, broad-angle, gradient-index nanostructure coating on the inside. For scale, the hole distance in the breadboard below is 25 mm.

windows feature an extremely broadband (reflectivity < 0.5 % over several hundred nanometres) and broad-angle (0° to > 45°) gradient-index antireflection nanostructure coating.[12] On the outside, the windows are uncoated; this combination offers maximum flexibility with respect to the wavelengths that can be transmitted into or out of the cell while still allowing external cleaning of the cell from dust, etc.

One small- and one large-diameter quartz glass tube with polished flat end lips (Fig. 5) are attached to the bottom and to one of the side borings of the quartz glass corpus, respectively. The bottom tube allows to insert the microscope objective and its mount (Fig. 3) into the quartz glass cell. The side tube connects the cell to the existing UHV apparatus and forms the single support of the cell against gravity, whereby we avoid static overdetermination which could lead to stress peaks and breaking.

## 3.2  Mounting concept

During the microscope assembly, the objective in its mount, sitting on the bottom fused-silica window, was carefully inserted into the quartz glass cell from below using a scissor jack, until a glass-to-glass indium seal could be formed (see below). To connect the quartz glass side tube to the tube cross, we engineered a small, steel flat-to-knife-edge adapter piece. In a first, critical step, the flat face of this adapter piece (cf. Fig. 3 in Ref. [93]) was connected to the quartz glass cell using indium sealing. In a second, uncritical step, the knife-edge face of the adapter was connected to the steel cross (Fig. 6) using a standard copper gasket and con-flat (CF) steel flange. To form our indium seals in a controlled fashion and to protect them afterwards, we designed clamping rings that could be assembled around the quartz glass cell tube lips. These clamps were in-house machined from a high-performance polymer.[13] Screw holes arranged circularly around the clamps allowed to press the respective sealing

---

[12]RAR.L2, Tel Aztec LLC, MA/USA.
[13]PAS-PEEK GF30, glass-fibre reinforced polyether ether ketone, Faigle GmbH, Austria.

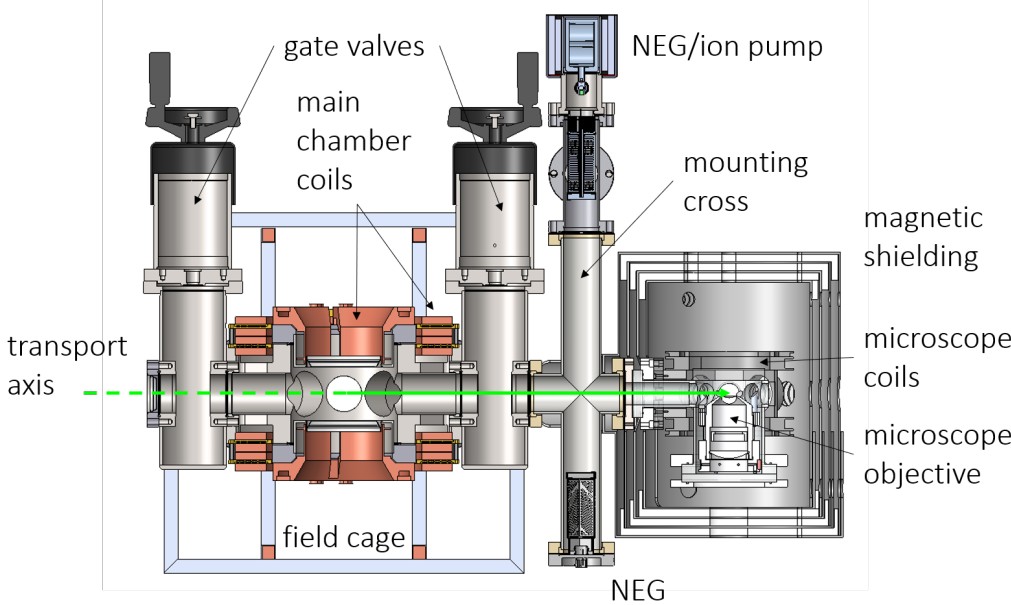

Figure 6: *Vacuum connection from main chamber* (left) *via steel cross* (centre) *to quartz glass cell* (right). The optical transport axis for the atoms is indicated in green. The magnetic shielding around the quartz glass cell is also shown. NEG = non-evaporable getter.

surface together and squash the indium O-ring in between. Conical beryllium-copper disc-springs, placed head-to-head under each screw, facilitated the loading of the clamps with even forces. The horizontal arms of the custom steel cross connect the quartz glass cell to the experimental main chamber via a UHV gate valve[14] and form part of the transport distance for our atomic samples. The bottom vertical arm connects to a non-evaporable getter (NEG) element,[15] whereas the top arm connects to (i) a combined NEG/ion pump module,[16] (ii) an ionisation gauge,[17] and (iii) an all-metal angle valve[14] for attachment of external mechanical pumps. All metal vacuum pieces were electropolished and vacuum-glowed[18] prior to assembly for reduction of $H_2$ outgassing as well as for reduction of magnetic permeability.

For microscope experiments, quantum gas samples produced in the Er-Dy apparatus main chamber are loaded into a single-beam optical dipole trap (ODT, 532 nm, around 15 W).[19] The ODT focus is then shifted through the vacuum connection into the microscope cell by translation of a relay lens on an air-bearing linear stage[20] (see Fig. 6).

### 3.3  Indium sealing

Forming a glass-to-metal connection for UHV applications is not trivial. For example, braze-alloy seals – as used by the majority of commercial manufacturers – require that the thermal expansion coefficients of the metal and the glass do not differ too strongly; otherwise temperature changes (which are not completely avoidable during the production process or bake-out) could lead to cracking of the glass.

Therefore, e.g., connecting stainless steel to quartz glass as in our case would require to form a gradient-index transition [102], i.e., to use several different glasses between the two end materials to gradually match the expansion coefficients. However, gradient-index transitions are typically long (10 to 20 cm) and mechanically weak.

In our design process, two concerns disfavoured a gradient-index transition. First, the prolongation of the atom transport distance; second, the risk of breaking when the cell and microscope are supported solely by a long, fragile gradient-index transition. We therefore decided to produce a direct quartz-glass-to-steel connection via indium sealing [103–105]. We also used this indium sealing procedure to form a glass-to-glass connection between our glass cell bottom tube and the bottom window once the objective was inserted. For us, indium sealing offered two main advantages. First, compared to adhesive glues, indium yields superior vacuum quality. Second, compared to, e.g., heat-diffusion bonding, it can be formed at room temperature – which is crucial since our objective may not be heated to above 90 °C [106].

Indium is a soft metal with a low melting point (about 156 °C [107]), low permeability, and low outgassing rates [108]. Exposed to air it is covered by a thin, passivating oxide layer. When mechanically deformed, however, like when pressed onto a glass surface, this oxide layer breaks and fresh, reactive metal is exposed. Such sticky, activated indium wets and reacts with glass, forming a tight seal. In the following, we outline some cornerstones of our indium sealing procedure.

*Indium material.* We used round indium wire,[21] activated in hydrochloric acid (37 %) for around 1 min straight before use, and connected two freshly cut, angled ends to an O-ring.

*Surface finish.* Our metal contact surfaces were lathed (not milled), with the stroke direction concentric with the indium O-ring; our quartz glass contact surfaces were polished

---

[14]VAT Vakuumventile AG, Switzerland.

[15]Capacitorr Z200, SAES Getters S.p.A., Italy.

[16]Nextorr D200, SAES Getters S.p.A., Italy.

[17]Tungsten-filament Bayard-Alpert gauge, Agilent Technologies, Inc., CA/USA.

[18]Reuter Technologie GmbH, Germany.

[19]Verdi V18, Coherent Corp., PA/USA.

[20]ABL 1500, Aerotech, Inc., PA/USA.

[21]Ø 0.05" ≈ 1.3 mm, In 99.995 %, Indium Corporation of America Co., MD/USA.

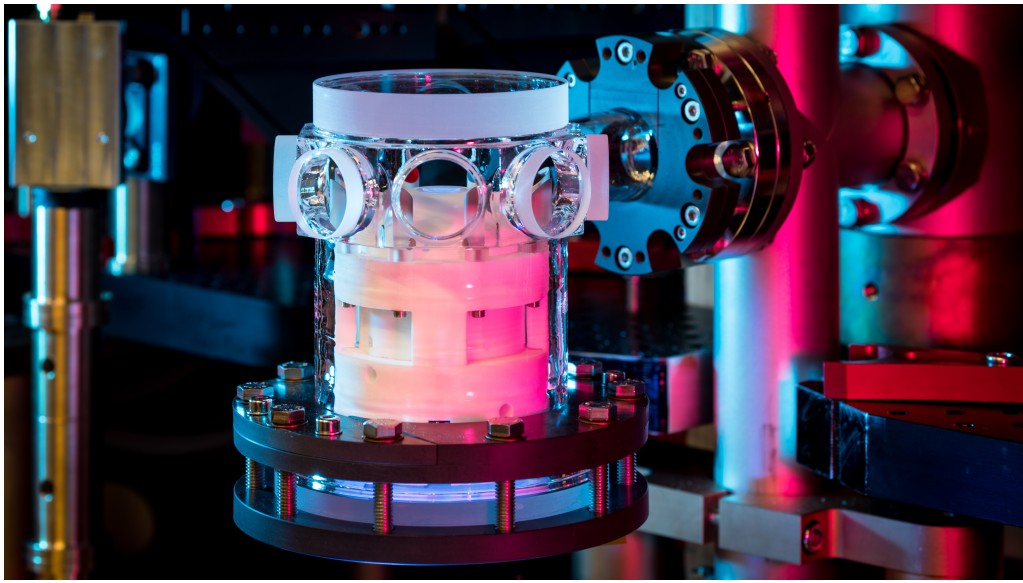

Figure 7: *The quartz glass cell with objective under UHV after attachment to the main apparatus.*

(optically clear, not matt) and remained uncoated in the sealing area.

*Surface cleaning.* Polluted surfaces underwent usual UHV cleaning: (i) cleansed in water and detergent, (ii) rinsed with distilled water, (iii) blow dried, (iv) acetone-cleaned in ultrasonic bath, (v) rinsed with fresh acetone, (vi) air-dried.

*Seal formation.* We gently and evenly squashed each indium O-ring between the two respective, clean contact surfaces by alternate tightening of screws around the clamping ring (visible on right and bottom of Fig. 7); a feeler gauge can help to monitor this process. We tightened the screws up to a few Nm torque, accompanied by careful visual inspection. After tightening of the screws, the indium seals typically had a thickness of around 0.3 mm. Thereafter we performed helium leak tests; small leaks may be closed by (i) stronger squashing, (ii) warming up of the seal region, or (iii) simply leaving the indium flowing for some hours [109].

### 3.4 Baking and attachment

When no more helium leaks were detected, our microscope assembly (still detached from the main experiment) underwent a gentle bake-out. For this, the quartz glass cell and the connection to the steel cross were enclosed in a clean metal container with resistive heating pads on the inside and insulation foam mats on the outside. The less sensitive parts of the steel cross were wrapped in layers of aluminium foil, heating wire, and insulating foam mats. Several thermocouples were used for temperature monitoring and to avoid gradients larger than a few degrees celsius over the whole assembly. We linearly increased the temperature by a few degrees celsius per hour up to 90 °C, then kept this temperature while monitoring the vacuum on a residual gas analyser.[22] After about two weeks, all relevant gas traces (in particular, $H_2O$) had fallen by several orders of magnitude and levelled off. We then linearly ramped down to room temperature by a few degrees celsius per hour. Note that the disk springs in our clamping rings serve a double purpose during the bake-out: first, they can take up force peaks when parts expand; second, they maintain the force when the indium softens and flows. After the bake-out, clamping screws were re-tightened to about the same torque as before; later, no more tightening was necessary.

---

[22]Stanford Research Systems, Inc., CA/USA.

After the bake-out was completed, the microscope assembly was flooded with dry argon gas, moved on its breadboard to our experiment table, and connected to the Er-Dy main chamber through a CF flange. After evacuation, no further baking was required. When the vacuum level as measured by our ionisation gauge had dropped to a level of around $10^{-11}$ millibar in the microscope section, the UHV gate valve to the Er-Dy main chamber was opened; lifetimes of quantum gas samples measured in the main chamber (around 40 s) remained unaffected by this. Figure 7 shows a photograph of the quartz glass cell with objective, after attachment to the Er-Dy apparatus.

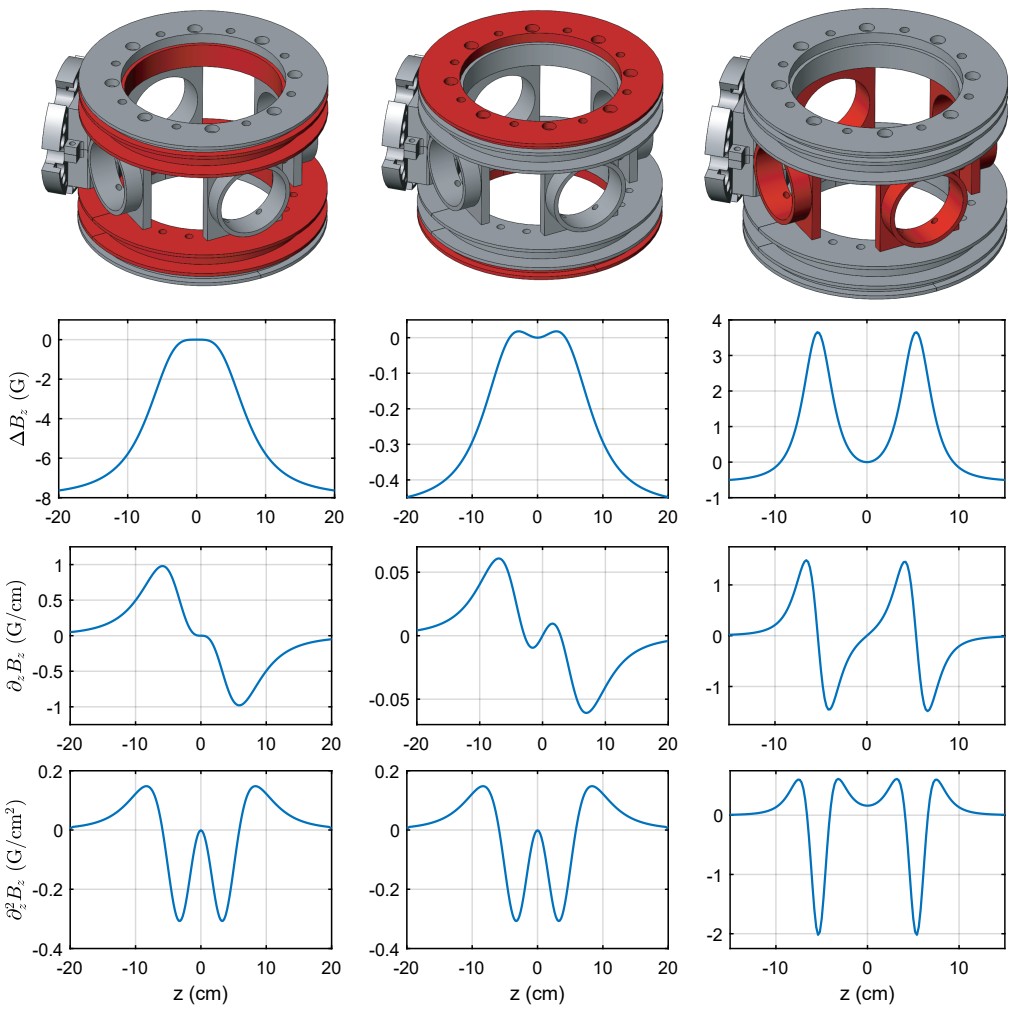

Figure 8: *Magnetic field coil system for the microscope chamber.* The fields are calculated by direct integration of the Biot–Savart law for 1 A of current, respectively. Note that in this figure we label the respective *local coil symmetry axis* by $z$. Slow vertical bias coils (*left column*), fast vertical bias or RF coils (*middle column*) and horizontal coils (*right column*). *First row*: Coil geometry. *Second row*: Field on axis (note that we plot the difference with respect to the atom position). *Third row*: Field gradient on axis. *Fourth row*: Field curvature on axis.

Table 4: *Coil specifications for the microscope cell.*

|  | VERT. SLOW | VERT. FAST | HORZ. |
|---|---|---|---|
| $r_0$ (mm) | 59 | 64 | 22.5 |
| coil separation (mm) | 54 | 78 | 98 |
| windings | 56 | 4 | 16 |
| centre field (G/A) | 8.0 | 0.5 | 0.6 |
| inductance[a] (F) | $3 \times 10^{-5}$ | $2 \times 10^{-6}$ | $3 \times 10^{-6}$ |
| resistance[b] (m$\Omega$) | 700 | 55 | 80 |

[a] Analytical approximation for ideal Helmholtz pairs.

[b] For 1-mm$^2$ copper wire.

## 4 Magnetic environment

In all ultracold-atom experiments the ability to set the magnetic field in a precise manner is absolutely essential, to define quantisation axes, to control level splittings, or to tune s-wave interactions at Feshbach resonances. For magnetic atoms such as erbium and dysprosium, where magnitude as well as direction of the magnetic field are of decisive importance for the stability and behaviour of a sample, this is even more the case.

In the following, we describe our coil system for shaping magnetic fields in the quartz glass cell (Section 4.1), as well as a ferromagnetic enclosure we have designed to shield the atoms from external magnetic fields and noise (Section 4.2).

### 4.1 Microscope coils

The trade-off in the design process of the microscope cell coils was between achieving a maximum flexibility in the shaping and switching of magnetic fields, blocking a minimum of optical access, as well as constraints posed by our magnetic shield (spatially and in terms of material magnetisation).

Our coil system consists of four pairs of coils which are held in place by a mechanical support structure. The individual pieces of the support structure are CNC-milled from a high-performance polymer,[23] assembled around the quartz glass cell, and mounted directly to the cell flange. Two pairs of coils (slow vs fast) are along the vertical (i.e., gravity) direction ($z$), close to Helmholtz configuration; see Fig. 8. In contrast to the slow (high-field) pair of vertical bias coils, the fast vertical coil pair has only few windings and is intended for fast jumps in magnetic fields and for generation of RF radiation. Two identical, mutually orthogonal horizontal coil pairs are arranged along the diagonals of the transport direction (cf. Fig. 8) and allow to set the field in the $(x, y)$-plane. All coils except the two bottom vertical coils could be wound prior to assembly.

Even though FEM simulations (Section 4.3) indicate that the magnetisation threshold of the innermost layer of our magnetic shielding will not be reached up to fields corresponding to more than a hundred Gauss in the cell centre, in order to avoid magnetisation effects it will be advisable to restrict the fields at the atom location to the few-Gauss or low tens-of-Gauss level. Table 4 summarises the characteristics of the coils, Fig. 8 shows the parts of the coil design as well as the corresponding calculated fields, gradients and curvatures.

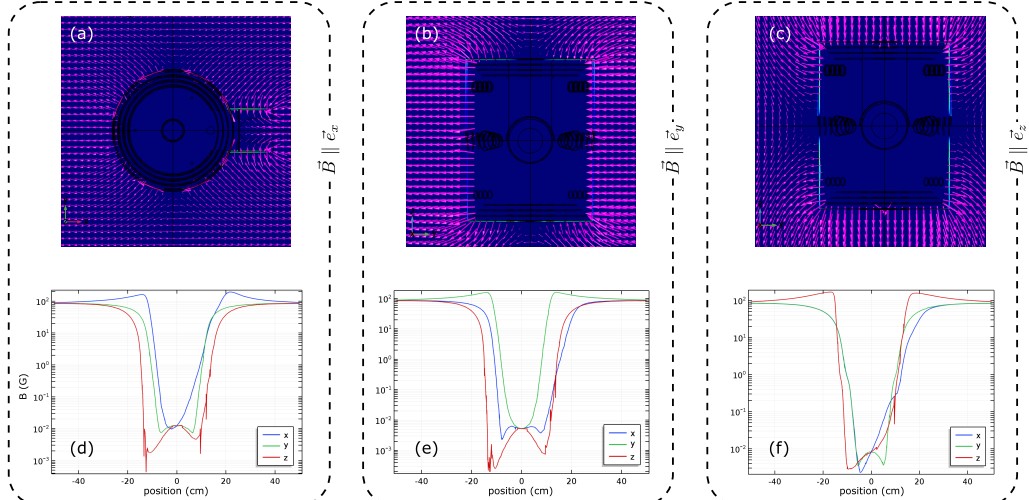

Figure 9: *FEM simulation of shielding efficiency.* (a–c) Vector fields (magenta arrows) of $\vec{B}$ when an external homogeneous magnetic fields is applied along the one of the three spatial directions, respectively. $\vec{e}_x$ (a, d) is along the transport axis, $\vec{e}_z$ (c, f) is along the cylinder axis. The strength of magnetic flux inside the metal is colour-coded, increasing from dark blue to red. (d–f) Calculated magnetic flux density along the $x$-, $y$-, and $z$-axis, plotted for the same external fields as in the top row (note that all three axes pass through shield openings). Our FEM simulations were carried out using the Comsol Multiphysics 5.4 software.

## 4.2 Magnetic shielding guidelines

For very field-sensitive measurements it becomes necessary to protect the atomic sample from magnetic stray fields present in the laboratory, such as the earth magnetic field or fields created by electric instrumentation. The protection strategy depends on the noise frequency.

Low-frequency (tens of Hz down to DC) noise can be reduced either by active compensation[24] or by enclosing the experimental chamber in a passive shielding. There are two types of passive magnetic shields, (i) superconducting shields at cryogenic temperature, and (ii) soft-ferromagnetic shields. Since ferromagnetic shields work at room temperature, they are much easier to integrate into quantum gas experiments [110–114]. The working principle of ferromagnetic shields is flux shunting, i.e., the shield has a high relative permeability and thus 'guides' the field lines around the protected volume. Among the most commonly used soft ferromagnetic materials are Mu-metal and Supra-50. Of these two, Mu-metal has the higher relative permeability and Supra-50 has the higher saturation flux density (see Table 5).

For fast oscillating fields (tens of Hz and higher), eddy current cancellation ('skin effect') becomes the dominant shielding process. This effect is the strongest for good conductors such as copper, but in practice also ferromagnetic DC shields typically provide a sufficient AC shielding [115]. The crucial task is therefore to find a good shielding for slowly-varying fields. For the Er-Dy experiment, we designed a passive multilayer ferromagnetic shield which will be surrounded by an additional, active field stabilisation system.

The performance of a magnetic shield is characterised by the shielding factor

$$S = B'/B \,, \tag{1}$$

---

[23]PAS-PEEK GF30, glass-fibre reinforced polyether ether ketone, Faigle GmbH, Austria.

[24]Since no probe for feedback can be put locally into the vacuum chamber, this is often realised using feed-forward.

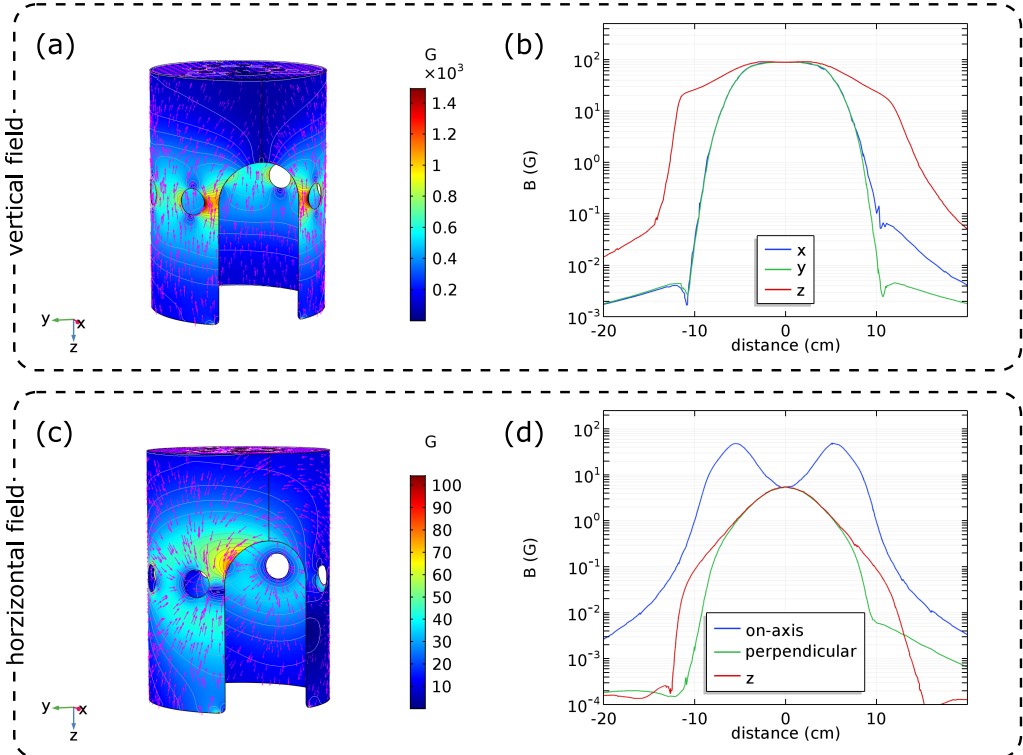

Figure 10: *FEM simulation of microscope coil effects. Left column* (a, c) shows *B* on the innermost shield layer (colour code); small magenta arrows indicate flux direction. *Right column* (b, d) shows the flux density *B* along spatial axes $x, y, z$. The coordinate origin is set to the atomic sample location; note that all three axes pass through shield openings. *Top row* (a, b) is for 10 A of current in the slow vertical coil pair (along $\vec{e}_z$). *Bottom row* (c, d) is for 10 A in one of the (diagonal) horizontal coil pairs (direction $\vec{e}_+$, where $\vec{e}_\pm = (\vec{e}_x \pm \vec{e}_y)/\sqrt{2}$). In (d), the blue, green, and red line correspond to the flux density *B* along the coil axis ($\vec{e}_+$), along the perpendicular axis ($\vec{e}_-$), and along $z$, respectively. FEM simulations were carried out using Comsol Multiphysics 5.4.

where $B$ ($B'$) is the field at the centre point in presence (absence) of the shield. We were motivated to target a magnetic shielding factor $S \sim 10^3$ by the following argument. To make the DDI in the lattice the dominant energy scale, it needs to be compared to the energy difference between adjacent Zeeman levels, scaling for bosonic erbium and dysprosium as $1.63\, h \times$MHz/G and $1.74\, h \times$MHz/G, respectively, where $h$ is the Planck constant. Magnetic field fluctuations measured in our laboratory are typically on the order of 5 mG. Hence, a shielding factor $S \sim 10^3$ (i.e., fluctuations $\sim 5\,\mu$G) would correspond to $\sim 10\, h \times$Hz, compared to an expected DDI on the order of $100\, h \times$Hz [42, 52, 117].

Basic analytical estimates (see Supplementary Information) already provide important guidelines for designing a magnetic shield:

Table 5: *Relative permeability, $\mu_r$, and saturation flux density, $B_s$, for the Er-Dy shield materials* [116].

| MATERIAL | $\mu_r$ | $B_s$ (G) |
|---|---|---|
| Mu-metal | $4.7 \times 10^5$ | $0.75 \times 10^4$ |
| Supra-50 | $2 \times 10^5$ | $1.5 \times 10^4$ |

*Geometry.* The best shielding performance is obtained for a spherical shell, followed by a cylinder (intermediate) and box (inferior) [115]. Shields that 'approximate a sphere' (i.e., have a similar characteristic length along all three spatial directions) show better performance.

*Size.* At fixed wall thickness, smaller shields are superior (Eq. A.4).

*Multilayer shields.* Nesting multiple shields with thin walls is better than a single shield with a thick wall – 'it helps to shield the shielding' (Eq. A.7).

*Discontinuities.* Avoid discontinuities (cuts, improper welds, etc.) to ensure unhindered guiding of magnetic flux. If discontinuities are unavoidable (e.g., for an assembly of multiple parts), the parts should have sufficient overlap and mechanical contact.

*Holes.* Avoid openings – they lead to flux leakage. If an opening is unavoidable, it can help to add a collar (exponential vs cubic suppression; Eqs A.2–A.3).

### 4.3 Microscope shield design

Our guidelines discussed above led us to favour a compact multilayer cylindrical shield. In particular, 'compact' means that the shield fits only the cell and magnetic coils, such that, e.g., all optomechanical components need to be placed outside. In our design process, we first drafted a prototype design based on the shield outlined in Refs [112, 113]. This prototype, a four-shell shielding, was made to comply with all geometrical constraints posed by our experiment. Each shell consists of a bottom and a top half, allowing to assemble the shield around the microscope cell. Second, we meshed the detailed prototype CAD model – including holes for optical access, cables, screws, etc. – and numerically analysed it using a finite-element method (FEM). The simulations[25] allowed us to study and refine our design with respect to two major factors: passive shielding performance and magnetic saturation caused by the microscope coils inside.

*Shielding efficiency.* To study of the shielding efficiency, we placed our model into external, static, homogeneous magnetic fields pointing along the three spatial directions.[26] In our case, the two most important improvements upon this study were (i) to maximally downsize the hole for the vacuum connection and to give it a collar, and (ii) to conically reduce the hole diameters for the side windows from outside to inside (at constant NA). Shielding efficiency simulation data for the revised design is shown in Fig. 9.

*Avoiding saturation.* We simulated the effects of the microscope coils inside our magnetic shielding. In our case, the two most important improvements following this study were (iii) to change the material of the innermost shell from Mu-metal to Supra-50 and, wherever needed, (iv) rounding of edges as well as adjustment of hole patterns and hole diameters to reduce local flux focusing. Some results of the saturation analysis for the revised design are shown in Fig. 10.

Our revised shield design was manufactured by Magnetic Shields Ltd, UK (Fig. 11). All shells are held in position by nylon spacers. After fabrication, the shells underwent a heat treatment (4 hours at 1150 °C) for magnetic annealing. Experimental tests of the shielding performance were carried out by the manufacturer inside a magnetically shielded room (to suppress the earth magnetic field, which is around 0.5 G in Europe). Using 3-axis Helmholtz coil pairs, uniform magnetic fields (from 0 to 10 G, and DC to 1 kHz) were applied along the three spatial directions and the residual field inside the shield was measured using a Bartington Mag-13 fluxgate magnetometer. The reported shielding factors are $> 10^3$ in both axial and transverse direction, meeting our initial target value. Once the shield and quantum gas samples inside the quartz glass cell will be ready, using the atoms as a magnetic field probe will allow a detailed in-situ study of the residual magnetic field noise.

---

[25] Comsol Multiphysics 5.4.

[26] Note that for a more conservative simulation of the shielding performance it could be helpful to artificially add small air gaps between shield pieces. In this way one takes into account that – due to manufacturing tolerances – pieces are typically not perfectly flush, which reduces the flux guiding.

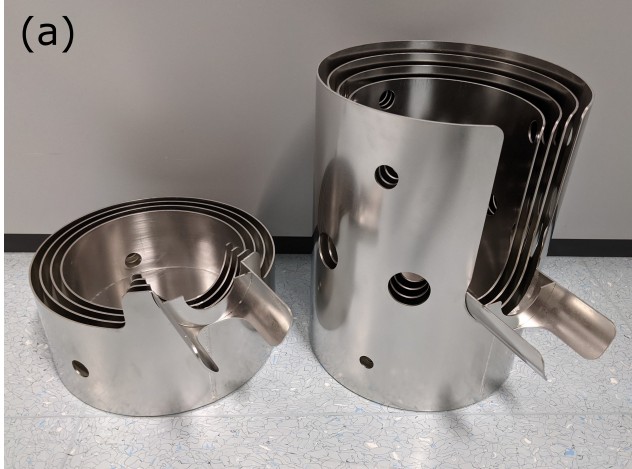
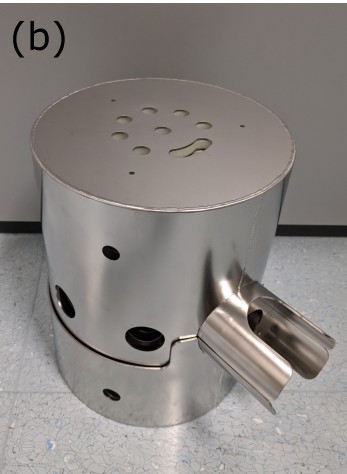

Figure 11: *Four-shell ferromagnetic shield.* (a) Bottom (*left*) and top half (*right, flipped upside-down*) of the magnetic shield. (b) The fully assembled shield. The collar on the side increases the shielding efficiency over the opening for the vacuum connection along the transport direction.

## Summary & Outlook

To conclude, we have designed and constructed a quantum gas microscopy apparatus for the highly magnetic lanthanoid elements erbium and dysprosium. Our microscope objective is non-magnetic, non-conducting, and features a high NA as well as a millimetre-scale working distance. The objective is mounted in vacuum, inside a glue-free, nanotexture-coated quartz glass cell. This combination offers a unique flexibility in terms of laser wavelengths that can be delivered onto the atoms, from $< 380\,\text{nm}$ to over $1700\,\text{nm}$.

We have developed and tested procedures and mechanical part designs for the formation of compact glass-glass and glass-metal (in particular: quartz glass to stainless steel) UHV seals using indium wire, which might inspire new routes in UHV apparatus design.

In terms of magnetic field control, we have designed, simulated and manufactured a versatile coil system which allows an accurate tuning of magnetic field in all spatial directions but impacts the optical access as little as possible. Further, we have designed and optimised a compact, four-layer ferromagnetic magnetic shield which can be assembled around quartz glass cell and coils. The shield suppresses external magnetic field noise by a factor of $> 10^3$ in all spatial directions and will facilitate conducting high-precision measurements.

Work towards getting our setup ready for dipolar lattice experiments is currently underway. Important next steps include (i) finalising the implementation of optical lattices, (ii) lattice loading from the transport optical dipole trap, (iii) implementation of the fluorescence excitation and detection systems as well as (iv) the development of – possibly lattice- and/or cooling-free – imaging procedures. Once fully operational, we believe that our microscope will make important contributions to the understanding of the complex physics of dipolar quantum many-body systems.

## Acknowledgments

We would like to thank all past and present members of the Er-Dy team, especially Philipp Ilzhöfer, Gianmaria Durastante, Arno Trautmann, Claudia Politi, Matthew A. Norcia, Lauritz Klaus, and Eva Casotti. We thank our group's Erbium, Tweezer, and Theory team, as well as

the whole Innsbruck Ultracold Atoms community. We are particularly indebted to Emil Kirilov, Innsbruck, for valuable discussions during the development process of indium handling and sealing procedures. We thank Aaron Krahn, Anne Hébert, Gregory A. Phelps, Sepehr Ebadi, and Susannah Dickerson from the Harvard Erbium team for fruitful exchange. MS is particularly grateful for an insightful research stay at Harvard. We acknowledge Dimitrios Trypogeorgos, INO-CNR and Università di Trento, for helpful discussions during the early design process of our magnetic shielding. We would further like to thank the IQOQI mechanical workshop team for their expert advice and manufacturing of many pieces necessary for the construction of our microscope setup.

**Author contributions**   This work is based on a chapter of M.S.'s dissertation, presented to the University of Innsbruck in July 2021 [118]. M.S. designed, simulated, assembled, and tested the microscope objective, UHV setup, and magnetic shield, with help from all team members. M.G. offered practical advice and general expertise during the design process of the microscope. M.J.M. and F.F. lead the Er-Dy experiment and supervised the development of the microscope apparatus. M.S. wrote the manuscript with valuable input from all other authors.

**Conflict of interest statement**   The authors declare that their research was conducted in the absence of any commercial or financial relationships that could be construed as a potential conflict of interest.

**Funding information**   This study received support from the European Research Council through the Advanced Grant DyMETEr (No. 101054500) and the QuantERA grant MAQS by the Austrian Science Fund (FWF, No. I4391-N). M.S. acknowledges funding from the FWF via DK-ALM (No. W1259-N27).

**Data availability statement**   The data sets presented in this work are willingly available upon reasonable request.

# A   Supplementary information

## Analytic approximations for ferromagnetic shields

In general, the shielding efficiency $S$ (Eq. 1) of a ferromagnetic shield depends on (i) the relative permeability $\mu_r$ of the shield material, (ii) the geometry of the shield, and (iii) the effect of potential holes.

### Shield geometries

Geometrically, the best shielding performance would be obtained for a spherical shell, however, for manufacturing reasons, the majority of magnetic shields has a cylinder (intermediate) or box form (inferior) [115].

### Effect of holes

*Flat surface.*   Consider an infinite plane with with surface normal along $\vec{e}_z$. Then a field $B$ entering through an opening in the plane will drop as [115]

$$B(z) \approx B(0) z^{-3} \,. \tag{A.1}$$

*Tube.* Consider a tube along $\vec{e}_z$ with diameter $D$. Then, magnetic fields entering through one end of the tube will decrease as [115]

$$B(z) \approx B(0)\,e^{-\beta z/D}\,, \tag{A.2}$$

$$\text{with} \quad \beta \approx \begin{cases} 7\,, & \text{for} \quad \vec{B} \perp \vec{e}_z\,, \\ 4.5\,, & \text{for} \quad \vec{B} \parallel \vec{e}_z\,, \end{cases} \tag{A.3}$$

i.e., shielding is better for transverse ($\perp$) than for axial ($\parallel$) fields.

**Cylindrical shields**

Let the shield be a hollow cylinder with $\mu_r \gg 1$, diameter $D$, length $L$, and shell thickness $d \ll D, L$.

We first consider the case $L \geq D$ and a homogeneous, transverse field. The flux entering through the ends is exponentially suppressed, hence, the residual field on the inside is strongly dominated by field "spilling" through the walls (cf. Fig. 1 in Ref. [119], e.g.). Therefore, such a cylinder can well be approximated by an infinite cylinder, for which the shielding efficiency has the analytic solution [120]

$$S^\perp = \frac{\mu_r d}{D}\,. \tag{A.4}$$

For axial fields, except for pathologic cases [119], the performance is improved when end caps are added to the cylinder. If they are taken into account, the axial shielding efficiency for a cylinder of size ratio $\alpha = L/D > 1$ can be described by

$$S^\parallel = \frac{4\eta S^\perp + 1}{1 + 1/(2\alpha)}\,, \tag{A.5}$$

$$\text{with} \quad \eta = \frac{1}{\alpha^2 - 1}\left(\frac{\alpha}{\sqrt{\alpha^2 - 1}}\ln(\alpha + \sqrt{\alpha^2 - 1}) - 1\right), \tag{A.6}$$

a geometry-dependent demagnetisation factor [120, 121]. For short cylinders ($\alpha \gtrsim 1$), the axial and transverse shielding efficiency are similar in magnitude, $S^\perp \gtrsim S^\parallel$. For long cylinders ($\alpha \gg 1$), however, the axial shielding loses effect ($S^\parallel \to 1$) due to field spilling through the walls. This demonstrates the advantage of cylindrical shields with $L \approx D$.

**Multi-layer shields**

The effect of a shield can be enhanced by enclosing it in a bigger shield. Following a magnetic circuit approach, the efficiency of a nesting of $N$ individual shields can be worked out analytically, with the dominant term [115, 122]

$$S \approx S_N \prod_{i=1}^{N-1} S_i \left(1 - \left(\frac{X_i}{X_{i+1}}\right)^k\right). \tag{A.7}$$

Here, $X \in \{D, L\}$ is the characteristic length scale of layer $i$, and we count the layers from inside to outside ($X_i < X_{i+1}$). $S \in \{S^\perp, S^\parallel\}$ is the shielding factor in the respective direction and – for cylindrical shields – given by Eqs A.4–A.5. The scaling exponent $k$ is geometry-dependent; in good approximation $k = 3$ for a spherical shield, and $k = 2$ and $k = 1$ for a cylindrical shield in transverse and axial direction, respectively [115].

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
