# Peer review of "A ship-in-a-bottle quantum gas microscope setup for magnetic mixtures"

_SciPost Physics, doi:SciPost Phys. 15, 182 (2023)_

## Round 1 · Referee Report · Anonymous (Referee 1) · 2023-7-17

Report

In 'A ship-in-a-bottle quantum gas microscope for magnetic mixtures' M. Sohmen and coauthors present a clean, well written and detailed description of the state of the art, the decision process, the building and the characterisation of a quantum gas microscope that is mounted inside the vacuum cell and externally shielded from magnetic fields by a passive mu-metal system. The paper introduction is well written, and the authors describe in detail the scientific reasons and the solutions that have been developed in the last 15 years to allow the detection of single atoms by using an optical microscope in ultracold gas experiments. The different solutions, their advantages and disadvantages are listed and catalogued in a very intuitive and useful way. The different approaches are then discussed with particular attention to their applicability in erbium and dysprosium experiments. The last part of the paper is dedicated to discussing the solutions that have been found for magnetic field control and shielding. Many technical details are really appreciated by experimentalist readers.

List of comments:

  • Introduction: B, paragraph 'DDI-mediated lattice'.  The authors present the interesting situation of a species dependent lattice. This is not a novelty in the field, and it has been discussed for spin mixtures as well. Citations are necessary. Examples of experiments are  Scattering in Mixed Dimensions with Ultracold Gases. G. Lamporesi et al, Phys. Rev. Lett. 104, 153202 Slow Thermalization between a Lattice and Free Bose Gas, David C. McKayet al Phys. Rev. Lett. 111, 063002 Theory citations are numerous and can also be found in the previous two papers.

The idea requires finding a wavelength suitable for the species-dependent lattice. Is there any idea or prediction for erbium and dysprosium? The spectra should be well known to give a rough idea of the location.

  • Microscope objective: Table II The table contains a very clear catalogue of advantages and disadvantages of the possible microscope solution. The citation list is, however, confusing and not clear to me. Are the citations chosen as exemplary or is there a completeness intention behind them? In this last case, many citations are missing, or a justification of the chosen citations must be provided. Missing citations, for example, Rev Sci Inst 91 063202 (2020) Rev Sci Inst 90 053201 (2019) Optics Express 28(24) 36122 (2020)

  • The 'Three ruby balls' sentence appears twice in different places with complementary information. Supposed to be so? 

  • III Magnetic environment: B. The motivations for the shielding and the requirements are well written. However, the citations 99–101 are relatively poor and incomplete compared to the completeness that characterises the part dedicated to the microscope.  I would suggest a more detailed state of the art. Especially to what it concerns the comparison with the shield system that is presented in 100. The four-layer system, the choice of material, and the geometry are practically identical to the one here presented, and more credits must be given.

  • III Magnetic environment: C. The authors report the experimentally measured magnetic shield factors in a very fast and not very detailed way. For example, a measurement as ref 100 would be much more satisfactory, if one compared how many experimental graphs are presented for the microscope.

  • validity: high
  • significance: high
  • originality: high
  • clarity: high
  • formatting: perfect
  • grammar: perfect

Author:  Maximilian Sohmen  on 2023-07-27  [id 3840]

(in reply to Report 1 on 2023-07-17)

We thank Referee 1 for his/her high rating of our work and the constructive feedback. We tried to incorporate the Referee’s points in a revised version of our manuscript.
This revised version will be made public at the end of the refereeing round (we would like to collect the suggestions from all referees). We thank Referee 1 for pointing us to these critical details, which in our opinion allowed us to improve our manuscript appreciably.

We address the Referee’s points here one by one:

**Introduction: B, paragraph 'DDI-mediated lattice'.**
We agree, a reference to works with non-dipolar gases was clearly needed. We thank the Referee for his/her suggestions; citations are now included.
Following the Referee’s suggestion, we now also give reference to works on spectroscopy and the polarizability of Er and Dy. These will be relevant for implementing a species-selective lattice. Since the spectra and, correspondingly, the polarizabilities are quite complex, there are several regions to be explored (e.g., in the range 400-800 nm) that come into question for application in a species-selective lattice. Which wavelength will work best is still to be determined and subject to future studies.

**Microscope objective: Table II**
We agree with the Referee, the list of references in Table II was subjective and incomplete. This table was indeed intended as a collection of prominent examples rather than a complete list. We added the references suggested by the Referee and corrected the table header to “Examples”.

**The 'Three ruby balls' sentence**
We thank the Reviewer for pointing us to this sentence. We now removed the redundant information and tried to phrase it more clearly.

**III Magnetic environment: B.**
We are thankful for this comment. We extended the list of relevant references for the state of the art. Further, we fully agree that we should have been more generous with acknowledging the results of the Trento group. We benefitted enormously from their work. We now tried to do them more justice and explicitly state that our design was based on theirs.

**III Magnetic environment: C.**
We fully agree with the Referee, details on the characterisation of the shield was clearly missing. We now explicitly include information on how the performance was determined (measurement by the manufacturer in on-site specialised facility). A characterisation of the magnetic field stability by means of atomic spectroscopy as by Farolfi et al. (2019) -- as suggested by the referee -- is planned as soon as our setup will be ready.

Anonymous on 2023-07-27  [id 3841]

(in reply to Maximilian Sohmen on 2023-07-27 [id 3840])

The Authors reply in a fully satisfactory way and I'm looking forward to see the new version at the end of the process.

---

## Round 1 · Referee Report · Anonymous (Referee 2) · 2023-7-20

Report

In their manuscript, Sohmen et al. present a well-written and detailed description of building and characterizing a quantum gas microscopy setup for probing erbium-dysprosium mixtures that is mounted inside the vacuum cell and shielded from magnetic fields. The introduction detailly summarizes the scientific developments of quantum gas microscopy during the past 15 years and motivates clearly that the mixture of erbium and dysprosium offers novel and exciting research with such a device. The authors detail the building and design process of the most important parts of the additional apparatus, i.e., the microscope objective, its vacuum integration, and the magnetic shielding, in three separate parts. Thereby, they consider different options and solutions with respective advantages and disadvantages in a very clear and understandable way and explain their design choices in detail. Moreover, they characterize crucial parts of the setup such as the optical performance and the magnetic shielding. In the last part of the manuscript, the features of the constructed quantum gas microscope are summarized and future steps are outlined. Due to the high number of technical details and the clear and detailed description of design possibilities and choices, the manuscript is a relevant contribution to the field of quantum gas microscopy and will be highly appreciated by experimentalists. However, I have a few questions and minor comments before I can recommend publication.

List of Comments:

General: - The referencing of figures and tables should be done in a consistent and ordered way. During reading, I found figures mentioned directly in the text as ‘Figure’ and ‘Fig.’, in ‘()’, in ‘[]’, with ‘cf. …’, with ‘see …’. Furthermore, figure 3(a) is mentioned before figure 2 and figure 7 before figure 6 in the main text. Moreover, I did not find a reference to figure 10 at all – I guess that the second time Fig. 9 is referenced Fig. 10 is meant to be referenced. All these issues are distracting while reading an otherwise very well-written and interesting manuscript.

Introduction, A: - ‘Owing to the superior signal-to-noise ratio,site-resolved detection is usually based on fluorescence imaging.’ What are other imaging techniques that fluorescence imaging is superior to?

Introduction, B: Being not an expert on erbium and dysprosium I have several questions concerning the features of the individual species as well as the mixture:

  • ’Most importantly, of course, erbium and dysprosium feature large permanent magnetic dipole moments of 7 μB and 10 μB, respectively, where μB is the Bohr magneton (cf. rubidium: 1 μB).’ Can the authors further comment on the consequences for DDI due to this one order of magnitude increased magnetic dipole moment compared to Rb?

  • ‘The erbium and dysprosium isotopes offer dense Feshbach spectra with a comfortable number of broad resonances at easily accessible field strengths, favourable for contact interaction tuning [51–53] or molecule formation [54].’ Having several control knobs at hand is in general good, but can such ‘dense’ Feshbach spectra lead to additional obstacles in contrast to other species? If yes, how can these be approached?

  • ‘Note that the broadest transitions of erbium and dysprosium are in the blue part of the visible spectrum, hence yield a high resolution according to the Abbe limit.’ I totally agree with this sentence, but I think it should be also mentioned that already in this regime a lot of standard optical elements widely used do not perform in this limit. E.g., lenses show a steep dispersion of the refractive index at these wavelengths.

  • $λ_l$ is tuned to a value where the polarizability vanishes for species A, $\alpha_A(λ_l$) ≈ 0, but not for species B.’ Citations for other mixtures fit here. Can the wavelength be specified/calculated or is there already a publication for the targeted mixture?

I. Microscope Objective, B: - ‘The objective’s optical design values are summarized in Table III; moreover, some important calculated characteristics are plotted in Fig. 2.’ How were these values and characteristics obtained? If provided by the manufacturer it should be mentioned, if obtained by optics simulation the program used should be mentioned.

I. Microscope Objective, C: - ‘Therefore we used numerical methods to design a telefocus system which consists solely of stock lenses, has a large effective focal length (6.2 m) but a small physical length (1.1 m) and is fully achromatic at 401 and 421 nm.’ This system’s design parameters and a few details could be included in an appendix. Has this system's "fully achromatic" behavior also been tested experimentally to rigorously rule out that the observed focal shift stems from the objective?

III. Magnetic environment, A: - ‘Even though FEM simulations (Section IIIC) indicate …’ Which program has been used in these simulations? From the figures shown, I would guess COMSOL, but which version and which module?

III. Magnetic environment, B:

  • ‘Estimated requirements for our future microscope experiments suggested to target a shielding factor S ∼ 10^3.’ How is this estimate carried out? What are the expected consequences and advantages for the experiment operated later?
  • validity: high
  • significance: high
  • originality: high
  • clarity: high
  • formatting: good
  • grammar: excellent

Author:  Maximilian Sohmen  on 2023-08-25  [id 3927]

(in reply to Report 2 on 2023-07-20)
Category:
remark
answer to question

We are very grateful for the referee’s critical reading, constructive comments, and positive rating of our work. This definitely helped us improve our manuscript. Below, we respond to the points of the referee in detail.

General

The referencing of figures and tables should be done in a consistent and ordered way. During reading, I found figures mentioned directly in the text as ‘Figure’ and ‘Fig.’, in ‘()’, in ‘[]’, with ‘cf. …’, with ‘see …’. Furthermore, figure 3(a) is mentioned before figure 2 and figure 7 before figure 6 in the main text. Moreover, I did not find a reference to figure 10 at all – I guess that the second time Fig. 9 is referenced Fig. 10 is meant to be referenced. All these issues are distracting while reading an otherwise very well-written and interesting manuscript.

We are impressed by and more than grateful for such a careful reading of our manuscript. Some of the points raised by the Referee here were definitely mistakes on our side, which we have tried to correct in the revised version. In particular: * The figures are now mentioned in the text in right order * Figure 10 is now mentioned in the main text – the referee is perfectly right, the second reference to Fig. 9 was intended to refer to Fig. 10.

On the other points raised by the referee, please allow us to comment: * When referring to figures in the main text, we follow the Physical Review Style and notation guide, which recommends “that the word figure is written out when it begins a sentence, but it is abbreviated at other times” (https://journals.aps.org/files/styleguide-pr.pdf, p. 11) * Also concerning the use of parentheses we follow the APS style guide: “Square brackets enclose a phrase that already contains parentheses.” (APS style guide, p. 14) * It is true that we use both, “cf.” and “see”. This is rather intentional, though. As recommended by many style guides (e.g. the Chicago Manual of Style, https://www.chicagomanualofstyle.org/home.html), “cf.” is [only] used to suggest a comparison, and the word “see" is used to point to a source of information.

We kept to these style guides for the arXiv version of our manuscript. For the final article, of course, this might need to be adapted to the specific journal style.

Introduction, A

‘Owing to the superior signal-to-noise ratio, site-resolved detection is usually based on fluorescence imaging.’ What are other imaging techniques that fluorescence imaging is superior to?

There are a couple of other imaging modes that are commonly used for imaging of (bulk) quantum gases, such as absorption imaging, and phase-contrast imaging. Specifically concerning quantum gas microscopy (i.e., imaging of single atoms on an optical lattice), experiments predominantly use fluorescence imaging, since it offers a much higher SNR than, e.g., absorption imaging. Faraday imaging has also been demonstrated for single atoms (https://journals.aps.org/pra/abstract/10.1103/PhysRevA.96.033610). We have added this information to the revised manuscript.

Introduction, B

Being not an expert on erbium and dysprosium I have several questions concerning the features of the individual species as well as the mixture. ’Most importantly, of course, erbium and dysprosium feature large permanent magnetic dipole moments of 7 μB and 10 μB, respectively, where μB is the Bohr magneton (cf. rubidium 1 μB).’ Can the authors further comment on the consequences for DDI due to this one order of magnitude increased magnetic dipole moment compared to Rb?

In regions of the Feshbach spectrum where the contact interaction is comparable to or smaller than the dipolar interaction, the physics of dipolar quantum gases differs strongly from non-dipolar gases. Also note that the magnetic moment enters the interaction squared, that makes about two orders of magnitude higher DDI than for the alkali case. Additionally, most dipolar effects are determined by the ratio between contact and dipole-dipole interaction (comparing dipolar length to scattering length). Here also the mass enters, making e.g. chromium much less dipolar than erbium. There is not enough room here to list all experimental differences between lanthanoids and alkalis, so we refer to the Review by Chomaz et al. (Reports on Progress in Physics, 86, 2023, DOI: 10.1088/1361-6633/aca814). Just to name some prominent examples: Bosonic dipolar quantum gases feature a roton mode, which can be used to produce dipolar supersolids. Fermionic dipolar quantum gases feature a deformation of the Fermi surface and show universal scattering, i.e., identical Fermions can scatter – this enables evaporative cooling without the need for a second species.

‘The erbium and dysprosium isotopes offer dense Feshbach spectra with a comfortable number of broad resonances at easily accessible field strengths, favourable for contact interaction tuning [51–53] or molecule formation [54].’ Having several control knobs at hand is in general good, but can such ‘dense’ Feshbach spectra lead to additional obstacles in contrast to other species If yes, how can these be approached?

The Referee is completely right, the dense Feshbach spectrum does not exclusively bring advantages, it also brings some challenges, which, however can be overcome. For example, continuous/exact tuning of the scattering length is not as straightforward as for alkali elements, since for Er and Dy there may be many (narrow) resonances to cross when ramping over a given magnetic field range. If such ramps or jumps over resonances are needed in the experimental sequence, atom heating and losses can become problematic – here in general it helps to jump fast or to use optical lattices, as demonstrated in our group with fermionic erbium. On the other hand, many useful tuning resonances occur at low magnetic field, where the strength of the applied field can be more easily controlled with high precision. These resonances are therefore routinely used for precise tuning of the interaction by us and other groups to successfully perform experiments with Er and Dy despite their dense Feshbach spectra. However, we would like to note that of course minimising B-Field noise is advantageous, especially when extremely low magnetic fields are desired. – This was also a motivation for the magnetic shield presented in our manuscript.

‘Note that the broadest transitions of erbium and dysprosium are in the blue part of the visible spectrum, hence yield a high resolution according to the Abbe limit.’ I totally agree with this sentence, but I think it should be also mentioned that already in this regime a lot of standard optical elements widely used do not perform in this limit. E.g., lenses show a steep dispersion of the refractive index at these wavelengths.

We completely agree with the Referee on this point. We might add that on top of the steep dispersion, around 400 nm many widespread glasses also show a non-negligible absorption. This is particularly critical for fluorescence imaging of single atoms, where only a small number (typically around 20) photons per atom is collected for localisation. The steep dispersion and reduced choice of (suitable) glasses were main challenges in the design of our in-vacuum objective. We have added a corresponding remark in the manuscript.

‘λl is tuned to a value where the polarizability vanishes for species A, αA(λl) ≈ 0, but not for species B.’ Citations for other mixtures fit here. Can the wavelength be specified calculated or is there already a publication for the targeted mixture?

This point was already raised by Referee 1. We have added references for non-dipolar mixtures to the revised manuscript. Several experimental works have investigated the polarizability of erbium and dysprosium (citations added) which suggest promising candidate regions for realising a species-selective lattice. However, which wavelength will ultimately work best is yet unclear and will be determined in future experiments.

I. Microscope Objective, B

‘The objective’s optical design values are summarized in Table III; moreover, some important calculated characteristics are plotted in Fig. 2.’ How were these values and characteristics obtained? If provided by the manufacturer it should be mentioned, if obtained by optics simulation the program used should be mentioned.

We thank the Referee for bringing this point to our attention. The calculated characteristics were simulated using Zemax Optic Studio 16. We have added a note in the manuscript.

I. Microscope Objective, C

‘Therefore we used numerical methods to design a telefocus system which consists solely of stock lenses, has a large effective focal length (6.2 m) but a small physical length (1.1 m) and is fully achromatic at 401 and 421 nm.’ This system’s design parameters and a few details could be included in an appendix. Has this system's fully achromatic behavior also been tested experimentally to rigorously rule out that the observed focal shift stems from the objective.

We agree with the referee, more information on this point will be interesting to the readers. Our telefocus design consists of a short-focal-length Hastings achromatic triplet in combination with a long-focal-length achromatic doublet (Thorlabs). We have not tested this telefocus system yet in combination with the microscope objective (now under vacuum). Once we will have atoms in the lattice, we will of course have to fully characterise our complete imaging system before images can be interpreted reliably. We have updated this information in the revised manuscript.

III. Magnetic environment, A

‘Even though FEM simulations (Section IIIC) indicate …’ Which program has been used in these simulations From the figures shown, I would guess COMSOL, but which version and which module?

The Referee is perfectly right, we used Comsol Multiphysics 5.4. We have added a note in the manuscript.

III. Magnetic environment, B

‘Estimated requirements for our future microscope experiments suggested to target a shielding factor S ∼ 10^3.’ How is this estimate carried out? What are the expected consequences and advantages for the experiment operated later?

This is a very good point. We can give a short order-of-magnitude estimate here. The typical magnetic field stability measured in our lab is by around 5 mG (i.e., S ~ 1000 would give around 5 uG stability inside the shielding). To make the dipole-dipole-interaction (DDI) in the lattice the highest energy scale, we need to compare it to the Zeeman shift. The difference between two Zeeman levels scales as ~ 1.63 MHz/G for bosonic Er and 1.74 MHz/G for bosonic Dy. Therefore, a field stability of 5 µG corresponds to ~8 Hz, compared to an expected DDI on the order of 100 Hz. We have updated the revised manuscript. Advantages for the experiment are numerous and important. In particular, the passive shielding will also help to reduce magnetic noise from the (lab) environment. As we have touched upon above, Er and Dy have a complex Feshbach spectrum with many narrow resonances. Our experiments (which typically crucially rely on a precise tuning of the scattering length) will benefit from the reduction of magnetic field fluctuations at the sample position.

---

## Round 1 · Referee Report · Anonymous (Referee 3) · 2023-8-8

Report

The manuscript by Sohmen et al. reports on a design study of elements required to realize a quantum gas microscope for dipolar atomic mixtures including assembly descriptions into a quantum gas apparatus. Microscopy of magnetic erbium atoms in an optical lattice has been demonstrated only recently, with spectacular demonstrations of strongly correlated phases emerging from long-range dipolar interactions (see arXiv:2306.00888). The manuscript thus accompanies these exciting advances in the field of cold atom research and provides interesting technical details associated with the construction of a quantum gas microscope for dipolar lattice experiments. Following a detailed introduction to past achievements and typical realizations of quantum gas microscopy and future research directions for dipolar mixtures, the core of the manuscript elaborates on the design of the in-vacuum objective and the magnetic shielding. Before I can recommend publication, the authors should address the following list of questions and comments:

- The recent achievement of quantum gas microscopy for dipolar atoms should be emphasized more prominently in the manuscript. A suitable paragraph for this would be §3 of the introduction. Also later in the manuscript, it is stated: “Many more already exist or might become apparent once a dipolar quantum gas microscope is in operation.” This should be corrected accordingly.
- Table I: The authors should provide more precise (literature) values for the linewidths of the listed transitions in Er and Dy.
- Introduction, section B: The large masses of Er and Dy are highlighted as key features providing small recoil energies. I agree that this may help e.g. for high-fidelity imaging. On the other hand, doesn’t the small recoil energy in a lattice system also necessitates lower temperature scales?
- Microscope objective design: For a technical paper, there is little information provided on the optical design of the objective. A figure indicating the lens assembly of the five singlets with focal lengths, geometries, and used materials would be very helpful. Additionally, the authors mention chromatic correction also at 626nm, but Fig. 2 and also Table III only specify simulated performance at 401 and 421nm. The plots and the table should be completed by the third wavelength. I suppose the plots in Fig. 2 and the values for Table III are extract from a Zeemax model provided by the manufacturer. This should be indicated. Finally, when Fig. 2 is referenced on p. 6, I am missing a discussion of the most central results of the objective analysis. For example, what do the authors extract in view of the performance of the microscope from panels (c) and (d) showing OPD and Modulus of OFT.
- On page 5, the authors describe a crucial element of the lattice architecture, namely the small mirror glued on the last lens element of the objective. Is this last lens surface flat? How is the alignment of this mirror done?
- On page 7, the authors mention an achromatic telefocus system with reduced physical lengths for imaging. It would be helpful to provide a sketch of that system or at least a more detailed description in the text.
- On page 7, the authors describe the quartz cell: which synthetic quarz glass is used? This would be valuable information specifically for UV applications.
- On page 9, the authors describe the bakeout and mention that no lifetime changes have been measured for the quantum gas after opening the gate valve to the microscope chamber. I suppose this is a statement for a gas in the main chamber? This should be indicated. Also, the value for the measured lifetime should be given. Does it confirm the 10^-11mbar, possibly measured on a gauge?
- Chapter III. The authors should provide a benchmark number for magnetic field stability they aim to achieve in their setup together with a more in-depth discussion of the associated physics they aim to observe.
- The manuscript does not mention water cooling for the microscope coils. What are the maximum field values that can be applied to this setup for typical duties cycles without major coil heating?
- The authors mention in the text that they have verified S~10^3 shielding factor at DC. What are simulated/measured AC shielding factors, and did the authors do a systematic measurement of the AC response of the shield?
- Paper title: The title of the paper suggests the realization of an operating quantum gas microscope tested with atoms. I would suggest to change the title in a way that clarifies that this manuscript is a design study and characterization of experimental components needed to perform quantum gas microscopy for magnetic mixtures.
  • validity: high
  • significance: high
  • originality: high
  • clarity: high
  • formatting: good
  • grammar: excellent

Author:  Maximilian Sohmen  on 2023-08-25  [id 3929]

(in reply to Report 3 on 2023-08-08)
Category:
remark
answer to question

We thank Referee 3 for his/her critical and constructive feedback on our work. We address below the points raised by the Referee. We have also tried to work these points into the revised version of the manuscript.

The recent achievement of quantum gas microscopy for dipolar atoms should be emphasized more prominently in the manuscript. A suitable paragraph for this would be §3 of the introduction. Also later in the manuscript, it is stated: “Many more already exist or might become apparent once a dipolar quantum gas microscope is in operation.” This should be corrected accordingly.

We fully agree with the Referee on this point. Let us note that the preprint mentioned by the Referee appeared only shortly before the submission of our manuscript, so during our writing phase the storyline was still a bit different than it is now. We have added direct reference and according sentences of clarification to the manuscript.

Table I: The authors should provide more precise (literature) values for the linewidths of the listed transitions in Er and Dy.

We fully support the point of the referee. We have improved the table in the revised manuscript.

Introduction, section B: The large masses of Er and Dy are highlighted as key features providing small recoil energies. I agree that this may help e.g. for high-fidelity imaging. On the other hand, doesn’t the small recoil energy in a lattice system also necessitates lower temperature scales?

The Referee is right, for reaching certain ground states requirements on temperature are challenging. From our point of view this is not directly due to the high mass but rather due to the scale of the DDI. As presented in https://doi.org/10.1103/PhysRevLett.120.243201, current state of the art to address these state preparation challenges is to use entropy redistribution.

Microscope objective design: For a technical paper, there is little information provided on the optical design of the objective. A figure indicating the lens assembly of the five singlets with focal lengths, geometries, and used materials would be very helpful. Additionally, the authors mention chromatic correction also at 626nm, but Fig. 2 and also Table III only specify simulated performance at 401 and 421nm. The plots and the table should be completed by the third wavelength. I suppose the plots in Fig. 2 and the values for Table III are extract from a Zeemax model provided by the manufacturer. This should be indicated. Finally, when Fig. 2 is referenced on p. 6, I am missing a discussion of the most central results of the objective analysis. For example, what do the authors extract in view of the performance of the microscope from panels (c) and (d) showing OPD and Modulus of OFT.

We fully agree with the Referee that a detailed technical sketch of the objective would be nice. We regret to say that this is not completely in our hands to decide, since the objective was developed in collaboration with the manufacturer. We respect that the exact objective design is therefore shared propriety of us and the manufacturer, which we cannot publish without their consent. Note that our objective is not an exception here: the designs of most objectives used in quantum gas microscopes are not publicly available. Researchers interested in a similar objective can, however, contact the manufacturer directly. Fig. 2 and Table 2 only specify the performance for 401 and 421 nm, since these broad transitions are the clear best candidates for fluorescence imaging and therefore most relevant for our purpose. This is why we concentrate on these wavelengths in our presentation. It was simply intended as a sidenote that the objective has been designed to work also at 633 nm, which is the alignment wavelength of the manufacturer. It is coincidence that the 626 nm Dysprosium line is close to this value, and hence imaging performance should be not too bad. At some later point, this might or might not become useful for future experiments. We did not measure the resolution at 626 nm since, unfortunately, the SNOM fibres used for our measurements transmit only light between 400 and 550 nm. Concerning Fig. 2, we are grateful for the hint; we now mention the main results of the objective numerical modelling in the main text.

On page 5, the authors describe a crucial element of the lattice architecture, namely the small mirror glued on the last lens element of the objective. Is this last lens surface flat? How is the alignment of this mirror done?

This is a very important point. The last (or first as we number it in our manuscript) lens surface is concave (now explicitly mentioned in manuscript). Together with coating requirements not trivial to bring to agreement (highly AR for imaging, highly reflective for lattice), this was the starting point for the glued-mirror idea. The small mirror is flat on both sides and glued to the concave lens surface. Alignment and glueing were carried out by the objective manufacturer (information now included in manuscript).

On page 7, the authors mention an achromatic telefocus system with reduced physical lengths for imaging. It would be helpful to provide a sketch of that system or at least a more detailed description in the text.

This important point was already raised by Referee 2. We have expanded the description in the revised manuscript.

On page 7, the authors describe the quartz cell: which synthetic quartz glass is used? This would be valuable information specifically for UV applications.

The non-optical parts of our cell (octagonal corpus and tubes) are fabricated from a standard quartz glass not specified in greater detail by the manufacturer. The important part is only that the expansion coefficient matches the optical parts of the cell. The optical parts of the cell are manufactured from (synthetic) fused silica. As commented by the Referee, this was important for us in view of transmission at 401 and 421 nm (imaging), as well as possible UV applications, and thermal lensing with respect to high-power lattice lasers.

On page 9, the authors describe the bakeout and mention that no lifetime changes have been measured for the quantum gas after opening the gate valve to the microscope chamber. I suppose this is a statement for a gas in the main chamber? This should be indicated. Also, the value for the measured lifetime should be given. Does it confirm the 10^-11mbar, possibly measured on a gauge?

We guess this is the sentence the Referee is referring to: “When the vacuum level in the microscope section had dropped to a level of around 10^−11 millibar, the UHV gate valve between microscope section and Er-Dy main chamber was opened; lifetimes of quantum gas samples measured in the main chamber remained unaffected by this.” We hope it is already clear that we refer to the lifetime in the main chamber. We have added the value for the lifetime (around 40 seconds).

Chapter III. The authors should provide a benchmark number for magnetic field stability they aim to achieve in their setup together with a more in-depth discussion of the associated physics they aim to observe.

We thank the Referee for this comment, which has connections to comments by the other Referees. We have added this information in the revised manuscript.

The manuscript does not mention water cooling for the microscope coils. What are the maximum field values that can be applied to this setup for typical duties cycles without major coil heating?

The referee is completely right, water cooling of coils is not planned. Since the coils will be in the magnetic shielding (which we do not want to magnetise) we will anyhow be limited to the few-Gauss level. At an experimental cycle time of around 10 s, of which maybe around 1 s will be spent in the glass cell, we are confident that heating of the coils will not pose a major problem.

The authors mention in the text that they have verified S~10^3 shielding factor at DC. What are simulated/measured AC shielding factors, and did the authors do a systematic measurement of the AC response of the shield?

This is a very important comment, which relates to comments by the other referees. We have complemented information on the shielding performance in the revised manuscript. Magnetic testing was carried out by the manufacturer in a specialised facility. The manufacturer states that shields are tested with external fields from 0 to 10 G and DC to 1 kHz. According to our test report, this revealed a shielding factor >1000 along all three spatial directions.

Paper title: The title of the paper suggests the realization of an operating quantum gas microscope tested with atoms. I would suggest to change the title in a way that clarifies that this manuscript is a design study and characterization of experimental components needed to perform quantum gas microscopy for magnetic mixtures.

We have rephrased the title: “A ship-in-a-bottle quantum gas microscope setup for magnetic mixtures”.

---

## Round 2 · Referee Report · Anonymous (Referee 2) · 2023-9-8

Report

The authors have clarified the manuscript and replied to all questions raised in a fully satisfactory way. I enjoyed reading the revised version and recommend publication.

---

## Round 2 · Author Response

Dear Editor, Dear Referees,
With gratitude and pleasure we submit the revised version of our manuscript.
We are much indebted to You all for the careful judgement and the fair and constructive feedback on our work. This has been a great help in improving several aspects the paper. We hope we succeeded in acknowledging all points duly.
If there are further points that deserve discussion, we are of course always available and ready to be contacted anytime.
On behalf of all co-authors,
Maximilian Sohmen

---

## Round 2 · List of Changes

In the revised manuscript, changes with respect to the earlier version are marked in red colour.
"Appendix" has been renamed into "Supplementary Information".

---

## Editorial Decision

published